# Resemblance of the global depth-distribution of internal-tide generation and cold-water coral occurrences

Anna-Selma van der Kaaden[1,2], Dick van Oevelen[1], Christian Mohn[3], Karline Soetaert[1], Max Rietkerk[2], Johan van de Koppel[1,4], Theo Gerkema[1]

[1]NIOZ Royal Netherlands Institute for Sea Research, Department of Estuarine and Delta Systems, PO Box 140, 4400 AC Yerseke, The Netherlands
[2]Copernicus Institute for Sustainable Development, Department of Environmental Sciences, Utrecht University, The Netherlands
[3]Department of Ecoscience, Aarhus University, Roskilde, Denmark
[4]Conservation Ecology Group, Groningen Institute for Evolutionary Life Sciences, University of Groningen, the Netherlands

*Correspondence to*: Anna van der Kaaden (annavanderkaaden@gmail.com)

**Abstract.** Internal tides are known to be an important source of mixing in the oceans, especially in the bottom boundary layer. The depth of internal-tide generation therefore seems important for benthic life and the formation of cold-water coral mounds, but internal-tide conversion is generally investigated in a depth-integrated sense. Using both idealized and realistic simulations on continental slopes, we found that the depth of internal-tide generation increases with increasing slope steepness and decreases with intensified shallow stratification. The depth of internal-tide generation also shows a typical latitudinal dependency, related to Coriolis effects. Using a global database of cold-water corals, we found that, especially in Northern Hemisphere autumn and winter, the global depth-pattern of internal-tide generation correlates ($r_{autumn}$=0.70, $r_{winter}$=0.65, $p$<0.01) to that of cold-water corals: shallowest near the poles and deepest around the equator with a decrease in depth around 25 degrees South and North and shallower north than south of the equator.

We further found that cold-water corals are significantly more often situated on a topography that is steeper than the internal-tide beam (i.e., where supercritical reflection of internal tides occurs) than can be expected from a random distribution: In our study, in 66.9% of all cases, cold-water corals occurred on a topography that is supercritical to the M2 tide, whereas globally only 9.4% of all topography is supercritical. Our findings underline internal-tide generation and the occurrence of supercritical reflection of internal tides as globally important for cold-water coral growth. The energetic dynamics associated to internal-tide generation and the supercritical reflection of internal tides likely increase the food supply towards the reefs in food-limited winter months. With climate change, stratification is expected to increase. Based on our results, this would decrease the depth of internal-tide generation, possibly creating new suitable habitat for cold-water corals shallower on continental slopes.

## 1. Introduction

The tide-generating force exerted by the sun and moon on the oceans causes long waves to travel over the ocean surface, called the barotropic tides. In the simplest case of a non-stratified ocean with flat topography, the water parcels move horizontally in

unison across the water column, along with a rise and fall of the ocean surface, typically at a diurnal or semidiurnal frequency. In reality, the ocean is stratified (layered) and has complex topography, giving rise to waves that travel in the interior of the ocean, as a movement of the isopycnals (levels of constant density). These baroclinic or 'internal' tides are an important

mechanism for mixing in the ocean (Vic et al., 2019; St. Laurent and Garrett, 2002; Garrett and Kunze, 2007). With amplitudes of up to hundreds of meters and associated turbulent cascades, internal tidal waves contribute to the redistribution of dissolved oxygen, organic matter, nutrients, heat, and salinity between the deeper and shallower ocean (Sarkar and Scotti, 2017; Jackson et al., 2012).

The internal tide thus increases the exchange of nutrients and organic matter between the deep and shallower layers of the
ocean, and by extension also increases benthic-pelagic coupling (Turnewitsch et al., 2016). Cold-water corals (CWCs) rely on organic matter that ultimately originates from primary production at the sea-surface (Van Engeland et al., 2019; Carlier et al., 2009; Maier et al., 2023). During its journey towards the deep-sea, organic matter is degraded by organisms in the water column, decreasing the food quantity and quality for benthic life at greater water depths (Snelgrove et al., 2018; Nakatsuka et al., 1997). Internal tidal dynamics can boost the vertical transport of organic matter towards the seafloor and thus facilitate
benthic life by increasing food availability (Soetaert et al., 2016; Vic et al., 2019; St. Laurent and Garrett, 2002).

Cold-water coral reefs are deep-sea ecosystems that have a particularly high organic-matter processing rate and have indeed been associated with internal (tidal) waves (e.g., Hanz et al., 2019; Roberts et al., 2021; Mohn et al., 2014; Davies et al., 2009; Wang et al., 2019; Juva et al., 2020; van Haren et al., 2014; Pearman et al., 2023). Cold-water coral reefs and mounds are built up of dead coral framework, coral rubble, and sediment, often with thriving CWC reefs on the mound tops (Roberts et al.,
2009). Using 6-hourly output from a 3D hydrostatic model, van der Kaaden et al. (2021) calculated the energy conversion rate (EC) from barotropic to baroclinic tides over a smoothed bathymetry of the Rockall Bank margin and found that the region of highest EC corresponds to the present-day location of CWC mounds. This suggests that the internal tide also plays a role in determining the region of coral mound initiation.

The depth-integrated and global total energy contained in the internal tide has been investigated before (e.g., Vic et al., 2019;
st. Laurent and Garrett, 2002; Yadidya and Rao, 2022), but the actual depth of internal-tide generation has not yet been considered in the present context and on a global scale. Yet, the depth of internal-tide generation is likely of relevance to CWCs, since it is this depth where the highest internal-wave excursions and more intense mixing occur near the seafloor (Mohn et al., 2014; van der Kaaden et al., 2021; Frederiksen et al., 1992).

Internal tides are generated over slopes. Their wave energy propagates away from the topography diagonally. So, internal tidal
waves travel away from the topography as a beam that makes an angle to the horizontal (Fig. 1). The steepness (i.e., the tangent of the angle with the horizontal) of the continental slope ($s$) compared to the steepness of the beam ($a$) is a key factor determining the strength of internal-tide generation. The bottom slope is critical when slope steepness equals the beam steepness ($s = a$); gentler bottom slopes are subcritical ($s < a$) and steeper slopes are supercritical ($s > a$). Since the wave angle of the M2 tide is rather gentle (3-8 degrees or a steepness of 0.01-0.14, i.e., 5-14 %) most continental slopes have a

region of near critical steepness for the M2 tide (Sarkar and Scotti, 2017), where intensification of beams occurs and where the generation is concentrated.

The beam steepness ($a$) is given by Eq. (1):

$$a = \sqrt{\frac{\omega^2 - f^2}{N^2 - \omega^2}} \tag{1}$$

with $\omega$ the tidal wave frequency in rad/s, i.e., $1.4052 \cdot 10^{-4}$ rad s$^{-1}$ for the semidiurnal M2 tidal constituent and $0.7292 \cdot 10^{-4}$ rad

s$^{-1}$ for the diurnal K1 tidal constituent. The beam steepness depends on the Coriolis frequency ($f$) and on how strongly gravity acts as a restoring force, i.e., the buoyancy frequency ($N$). The former is related to latitude: zero at the equator and increasing towards the poles. The latter is related to stratification: a steeper density gradient results in larger values for $N$. The depth and magnitude of the internal tide can thus be expected to vary with 1) continental slope steepness, 2) latitude, and 3) stratification. At latitudes where the Coriolis frequency exceeds the frequency of the tidal constituent, the wave energy in the internal tide

cannot propagate away from the topography. The internal tide is then said to be 'trapped' at the topography. Similarly as with 'critical slopes', near these 'critical latitudes' (around 75 degrees for the M2 tide and at 30 degrees for the K1 tide), strong currents and enhanced vertical mixing occur at the region of internal-tide generation (Pereira et al., 2002).

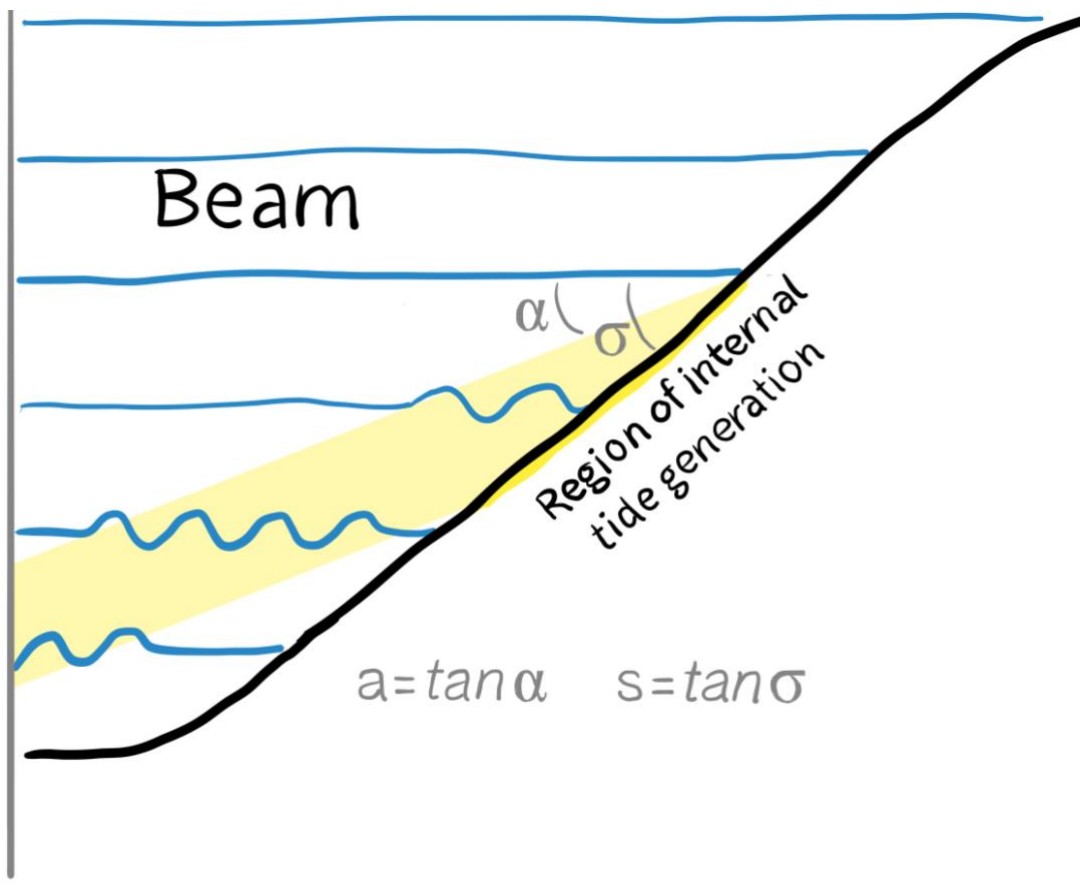

**Figure 1. Sketch of internal-tide generation on a continental slope. Internal tidal waves travel on the surface of isopycnals (blue lines). The energy in the internal tide (i.e., wave amplitude) travels away from the topography in the horizontal as well as in the vertical, forming a beam (yellow). The angle of the beam to the horizontal ($\alpha$) as compared to the angle of the continental slope ($\sigma$) is a parameter determining the depth and strength of internal-tide generation on the seafloor.**

Here, we investigate how the depth of the internal-tide generation on the continental slope changes along realistic transects extracted from around the globe. The energy conversion rate of barotropic to baroclinic tides is taken as a proxy for the strength of internal tides. We investigated the relationship between the depth of internal-tide generation on the continental slope and slope steepness, latitude, and stratification, with an idealized model setup and with realistic topography and buoyancy frequencies from transects. To study the importance of tidal dynamics for CWC reefs, we then compared the depth of internal-tide generation on the continental margin to occurrences of reef-building CWCs. This study contributes to the general understanding of the role of the internal tide for CWC communities.

## 2. Methods

### 2.1 Model description

Energy conversion from the barotropic to baroclinic tide (EC) was simulated with a linear hydrostatic internal-tide generation model. We assume uniformity in topography and all dynamic variables in the along-slope direction ($y$), making the model essentially 2D (except for a transverse velocity component $v$ that is induced by the Coriolis force). This approach can be justified since continental slopes vary mostly in the across-slope direction. A stream function can thus be introduced for the baroclinic cross-slope and vertical current speeds: $u = {\partial \psi}/{\partial z}$ and $w = -{\partial \psi}/{\partial x}$ respectively, resulting in the following linear hydrostatic model equations (Gerkema et al., 2004):

$$\frac{\partial^3 \psi}{\partial z^2 \partial t} - f\frac{\partial v}{\partial z} + \frac{\partial b}{\partial x} = 0, \tag{2}$$

$$\frac{\partial v}{\partial t} + f\frac{\partial \psi}{\partial z} = 0, \tag{3}$$

$$\frac{\partial b}{\partial t} - N^2 \frac{\partial \psi}{\partial x} = -\frac{z N^2 Q \sin \omega t}{[H - h(x)]^2}\frac{dh}{dx} \tag{4}$$

where $f$ is the Coriolis parameter and $b$ the buoyancy frequency expressed as "minus effective gravity" $b = g\frac{\rho}{\rho*}$ (m s$^{-2}$) with $\rho^*$ a constant representative value of density (kg m$^{-3}$) and $\rho$ the density perturbation with respect to the local static value. The right-hand side of Eq. (4) represents the forcing of the cross-slope barotropic tidal flux with amplitude $Q$ and semidiurnal (M2) tidal frequency $\omega$ (1.4052·10$^{-4}$ rad s$^{-1}$).

The bottom is described by $z = -H + h(x)$, where $H$ is the undisturbed ocean depth, $h(x)$ the topography, and a rigid lid surface is located at $z = 0$. The model was solved with a Chebychev collocation method using 60 Chebychev polynomials. In the vertical, we used 60 topography-following collocation points with increased resolution near the surface and bottom. In the horizontal direction and in time, a finite-difference method was used with steps of 0.4 km, and a temporal resolution of 1,000

time-steps per tidal period. A sponge layer of 150 km in the deep ocean and 50 km on the shelf dampened incoming waves

with a Rayleigh-friction term and a fourth-order spatial filter was applied to dampen fine-scale artificial oscillations. The model was forced with a barotropic cross-slope flux of 100 m$^2$ s$^{-1}$ for all simulations. We note that the strength of this forcing is immaterial to the spatial structure of conversion, as the model is linear. For further details on the numerical scheme, we refer to Gerkema et al. (2004).

The amount of energy that is converted from the barotropic into the baroclinic tide per second, per volume (W m$^{-3}$) is calculated

as:

$$EC = -\frac{\rho*}{T}\int_0^T dt\, b\, W \tag{5}$$

with $T$ one tidal period and $W$ the barotropic vertical velocity (m s$^{-1}$).

For the simulations, we focussed on the M2 tide, since the semidiurnal (or mixed semidiurnal) tide is dominant at most places regarding both surface elevations (Gerkema, 2019) and barotropic tidal current speed (Fig. 2 constructed using data from the

TPXO9 atlas).

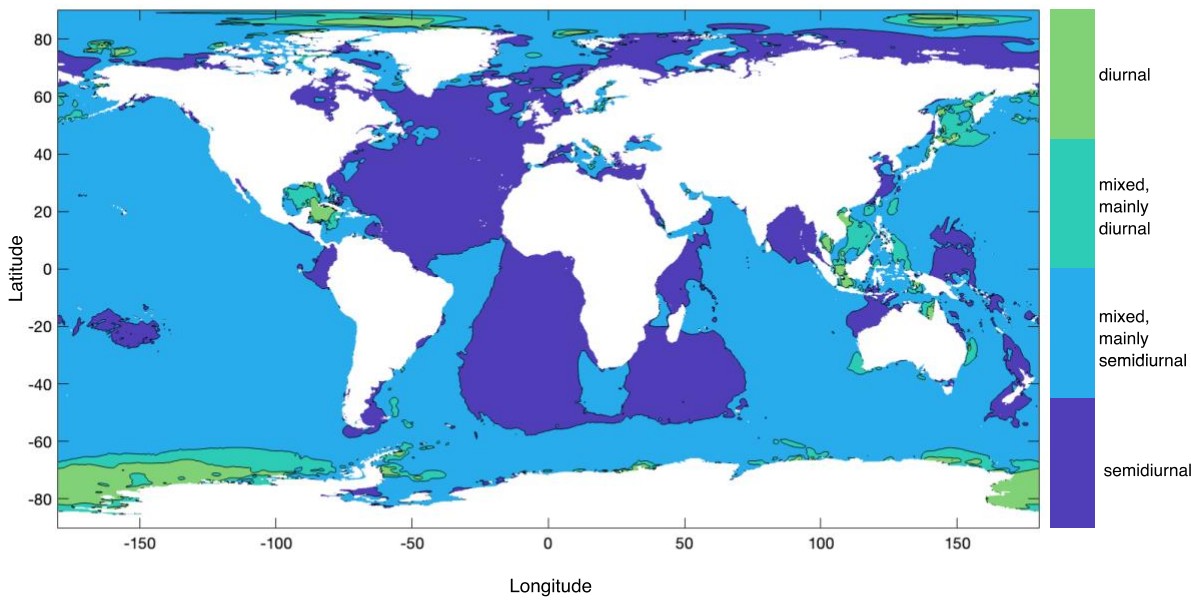

**Figure 2. Global distribution of semidiurnal (dark blue), diurnal (green) and mixed tides, based on barotropic tidal current amplitudes of the semidiurnal M2 and S2 tide and the diurnal K1 and O1 tide. The classification is based on the so-called form factor, as in** Gerkema (2019)**, and was calculated as the sum of amplitudes of the K1 and O1 constituents over the sum of those of M2**

**and S2 constituents. This map was created with the TPXO9 atlas** (Egbert and Erofeeva, 2002)**.**

## 2.2 Data

### 2.2.1 Bathymetry, stratification, and surface tides

Bathymetry for the global transects was extracted from the NOAA ETOPO1 global relief model (NOAA National Geophysical Data Center, 2009, accessed June 2022) with the *marmap* package in R (Pante and Simon-Bouhet, 2013). The topography has a spatial resolution of 1 arcminute, equivalent to a mean resolution of 1.89 km (0.71 km - 24.63 km, min - max).

Realistic stratification profiles were selected per season by calculating buoyancy frequency values ($N$) using salinity and temperature data from the Levitus seasonal dataset (Levitus, 1982). The Levitus database includes salinity and temperature in the ocean at 24 levels ranging from the surface down to 1500 m and was last updated May 2015. The seasonal dataset includes 4 seasons that are specified as: 1) February to April ('spring'), 2) May to July ('summer'), 3) August to October ('autumn'), 4) November to January ('winter'). Note that we classified the seasons with respect to the Northern Hemisphere (NH), so e.g., February to April is NH spring but Southern Hemisphere (SH) autumn.

Information on surface tides came from the global barotropic tidal model TPXO9-atlas v5 (Egbert and Erofeeva, 2002). Barotropic horizontal tidal current speeds are provided at 1/30 degree resolution.

### 2.2.2 Database of cold-water coral (CWC) occurrences

A global dataset of the occurrences of CWCs originated from the NOAA National Database for Deep-sea Corals and Sponges (NOAA National Database for Deep-Sea Corals and Sponges, version 20220426-0), ICES Vulnerable Marine Ecosystems (International Council for the Exploration of the Sea, June 2022) and OBIS (OBIS, 2022). From the NOAA database we selected all records of the main CWC reef-building species *Desmophyllum pertusum* (previously *Lophelia pertusa*), *Enallopsammia profunda*, *E. pusilla*, *E. rostrata*, *Goniocorella dumosa*, *Madrepora carolina*, *M. oculata*, and *Solenosmilia variabilis* (Freiwald et al., 2004; Maier et al., 2023), below 100 m depth, recorded from 1900 or later, with a horizontal location accuracy of 1,000 m or less, and between the critical latitudes for the M2 tide of 70 degrees North and South (15,629 records). From ICES VME, we selected all "Stony corals" VME indicators in VME habitat type "Cold-water coral reef" with a position accuracy of 1,000 m or less, and between 70 degrees North and South (379 records). From OBIS, we selected all records of the main reef-building species (see above), below 100 m depth, with a location accuracy of 1,000 m or less, and between 70 degrees North and South (26,117 records). These records are excluding all records where the stated depth was incongruous with the topographic dataset (i.e., exceeding the depth of the bathymetry or with a location on land), or with nonsensical latitude and longitude coordinates. Roberts and Cairns (2014) also mention the species *Bathelia candida* as a main CWC reef-building species. We did not include this species in our analysis, resulting in the omission of 4 eligible datapoints from the NOAA and/or OBIS databases near transect 7 (SH). The species might be included in the data from ICES, as we selected all "Stony corals" in "Cold-water coral reef" habitat.

We excluded all coral records marked as "dead" or "fossil". Our database contains 40,902 records in total. We used the ("middle") latitude and longitude coordinates and the mean depth. We calculated the local slope steepness at which CWCs are

found from ETOPO bathymetry at a 30 arc-minute resolution (similar as the smoothed model slope topography; next section 2.3.2) and at a 1 arc-minute resolution in R.


Cold water corals typically occur shallower near Norway, the Mediterranean Sea, the US west coast, and north Australia, and deeper in the open ocean (Atlantic and Pacific Ocean) than on continental slopes, except for Portugal and south of Australia where CWCs occur deep near the coast (Fig. 3). In some regions with CWC occurrences, mainly the Gulf of Mexico, the South China Sea, some parts of the Mediterranean Sea, and South of Australia, the M2 barotropic (surface) tidal current speed is <1

cm s$^{-1}$. In these areas the barotropic tidal signal is mainly diurnal (Fig. 2) and the influence of the internal M2 tide will therefore be limited. Hence, in our analyses, we only included those coral occurrences that are situated in regions where the barotropic tidal current is semidiurnal or mixed but mainly semidiurnal (as in Fig. 2).

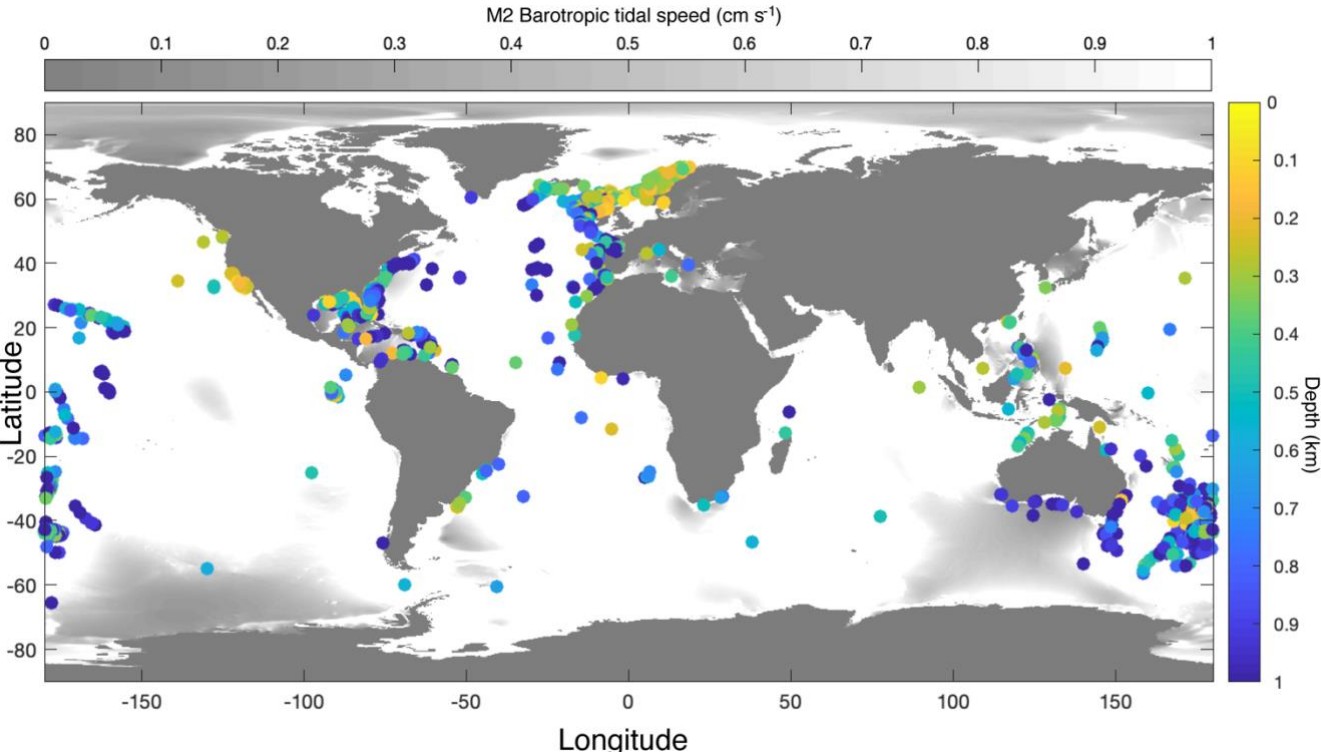

**Figure 3. Map with all coral occurrences from the databases with colour indicating the depth at which the corals were found. In the**
**oceans, grey shading indicates where the speed of the barotropic M2 tide is below 1 cm s-1. This map was created using the TPXO9 atlas** (Egbert and Erofeeva, 2002)**.**

### 2.3 Simulation settings

### 2.3.1 Idealized simulation setting

As a reference setting for the idealized simulations we chose a maximum slope steepness of 0.114 at 25 degrees latitude with
a vertical uniform stratification of 0.002 rad s$^{-1}$. The relationship between internal-tide generation and topographic slope was

investigated by running the model with various maximum topographic slopes, using the following values: 0.0228, 0.0285, 0.038, 0.057, 0.076, 0.114, 0.1425, 0.1899, 0.2279, and 0.2848. The idealized slope had a sigmoid shape. The relationship between internal-tide generation and latitude was investigated by running the model at latitudes of -70, -60, -50, etc. up to 70 degrees. The relationship between internal-tide generation and stratification was investigated with a vertically non-uniform

stratification ($N$). We interpolated $N$ from 0.001 rad s$^{-1}$ at the ocean surface to a value at 100 m depth that represents a thermocline. At 100 m depth, we varied $N$ from 0.001 rad s$^{-1}$ to 0.005 rad s$^{-1}$, at steps of 0.0005 rad s$^{-1}$. We interpolated $N$ from the value at 100 m depth to 0.001 rad s$^{-1}$ at 3 km depth. The $N$ -profile was smoothened before model simulation by cubic interpolation. All simulations were run for 20 tidal periods (starting from rest), after which the signal had become periodic over the slope.

**2.3.2 Realistic simulation setting**

For the realistic simulation setting, we selected transects on continental slopes from all ocean basins at every 10$^{th}$ latitudinal degree between 64.5 degrees South and 64.5 degrees North, resulting in 116 transects. The transects were drawn perpendicular to the coastline by hand. Since the model by Gerkema et al. (2004) assumes that a continental shelf is present, only transects starting from a well-defined continental shelf were included in the analysis. Transects where the tidal signal was not periodic

at the end of the simulations were excluded from the analysis, since the results on internal tide conversion rate would not be reliable in these cases. We thereby excluded 32 transects, resulting in 84 of the 116 transects being used in the analyses (Fig. 4).

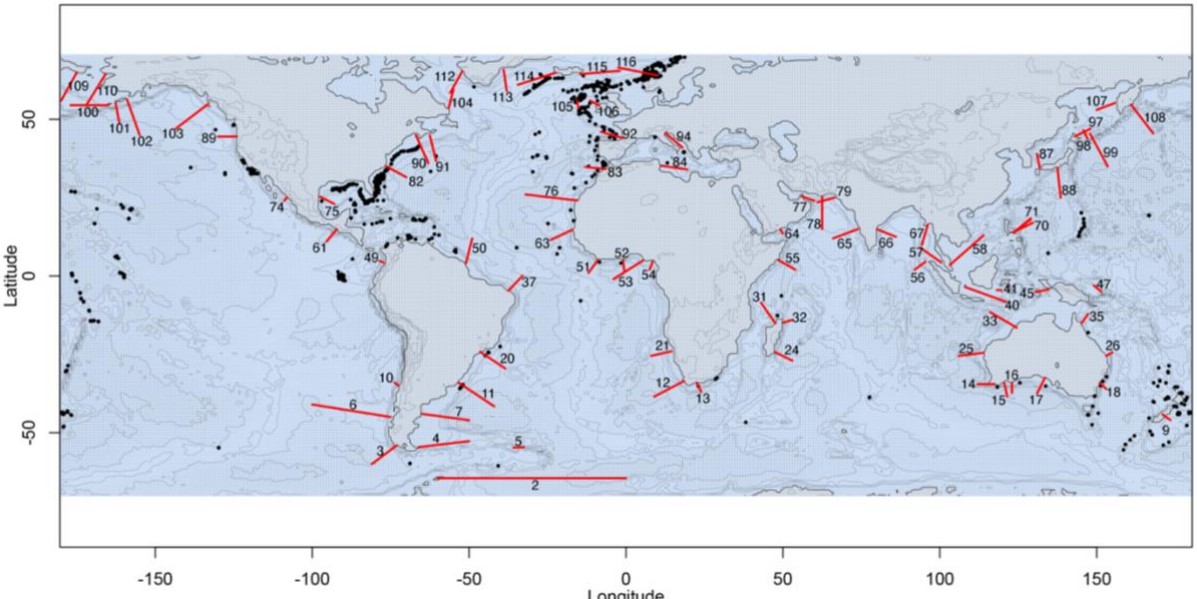

**Figure 4. Map of the 84 (out of 116) selected transects (red lines) on which internal-tide generation was simulated with realistic**
**topography and stratification. Some transects overlap with coral occurrences from the database (black dots). This map was created using the NOAA ETOPO1 global relief model** (NOAA National Geophysical Data Center, 2009)**.**

We investigate the general depth-pattern of internal-tide generation on continental slopes globally. To avoid interference and scattering by internal-tide generation over rough topography (e.g., abyssal hills), we smoothed the bathymetry by placing the mean value of every 30 points (30 arcminutes) in the middle of those 30 points. The smoothed model topography was then built by making a cubic interpolation between the remaining averaged points at 0.4 km resolution. Realistic stratification profiles used in the model were assumed horizontally uniform along the cross-slope transects. For the model setup we selected the stratification profile located at the deepest part of the transect and extrapolated $N$ from 5 m depth to the sea surface. For transects deeper than 1450 m depth, we set the value of $N$ at the maximum transect depth to $2 \cdot 10^{-4}$ rad s$^{-1}$. Our approach can be justified, because stratification typically changes very little below 1.5 km depth and is not well-mapped (Banyte et al., 2018). Also, while stratification at the boundaries of deep water-masses can cause internal wave generation, this happens typically over rough topography in the open ocean (Nikurashin and Ferrari, 2013; Banyte et al., 2018) whereas we focus on continental slopes. Cubic interpolation was used to define the buoyancy frequency at the collocation points of the model. The depth of the shallow and deep parts of the transects ranged from 0.1 m to 0.3 km and from 1.1 km to 8.9 km respectively.

We identified peaks in energy conversion rates (EC) at the model seafloor in all transects with Matlabs *findpeaks* function. Since we forced all transect simulations with the same barotropic flux, values of EC rates are not meant to be realistic and cannot be compared between transects; We are here only concerned with the structure of the EC field and in particular the depth of maximum conversion. We first scaled EC rates on every transect between 0 and 1 and identified all peaks with a minimum prominence of 0.05. We further calculated the effective depth ($\bar{z}$) of internal-tide generation weighted by EC rates as Eq. (6):

$$\bar{z} = \frac{1}{max(z) - \min(z)} \cdot \frac{\int_{min(z)}^{max(z)} z \cdot EC(z) dz}{\int_{min(z)}^{max(z)} EC(z) dz} \tag{6}$$

This way, the effective depth of generation really indicates where the bulk of the generation takes place (rather than being controlled by incidental narrow peaks in EC). Similarly, we calculated the weighted mean slope steepness and weighted mean latitude. We ignored incidental negative EC values, since negative values indicate a loss of energy from the baroclinic tide instead of a generation. See Fig. A1 (appendix A) for an example transect with EC peaks and weighted mean depth.

**2.4 Statistics**

We fitted a smoothing spline to the relationship between the depth of internal-tide generation and slope steepness, and to the relationship between the depth of internal-tide generation and latitude. Similarly, we fitted a smoothing spline to the relationship between the depth of CWC occurrences and slope steepness, and median depth of CWC occurrences and latitude. To evaluate the similarity between the depth-distribution of internal-tide generation and CWCs we calculated the Pearson correlation coefficients between the fitted curves of internal-tide generation and CWCs. We correlated the curves from a slope steepness of 0 to 0.6 and from -70 to 70 degrees latitude. Only corals situated within the region where the M2 tide is dominant (as in Fig. 2) were included in the analysis. We used the median depth of CWC occurrences to avoid effects of the global observation bias in CWCs (Davies and Guinotte, 2011).

We further calculated the proportion of regions on our simulated transects where the topography becomes supercritical, critical,
or subcritical for the internal M2 or K1 tide, or where the internal tide is trapped at the topography (i.e., poleward of 30 degrees for the K1 internal tide). We defined 'critical' as a region on the continental slope where the steepness of the topographic slope equals the steepness of the internal tide beam $\pm\ 5\cdot10^{-7}$. To compare whether internal tide peaks are more often than by chance found at regions where the topography is (super)critical, we calculated the same proportions for the regions where we found peaks in energy conversion rates. Similarly, to investigate whether CWCs occur more often than by chance on those regions
on the continental slopes that are (super)critical for the internal tide, we calculated the same proportions for CWC locations using global ETOPO bathymetry at a 30 arc-minute resolution (i.e., similar topographic resolution as used for the realistic simulations). These results obtained with a low-resolution bathymetry convey information about global trends. To get insight also into more regional processes, we calculated the same proportions for CWC locations using 1 arc-minute bathymetry and compared them to 10% of randomly selected sites from the global ocean. 95% confidence intervals for the proportions of
supercritical, critical, subcritical, and trapped internal tides were obtained from bootstrapping 1,000 repetitions. Two proportions are significantly different ($p<0.05$) when the 95% confidence intervals do not overlap. The steepness of the internal-tide beam depends on season, so these results are calculated using all four seasons separately.

## 3. Results

We investigated the relationship between the depth of internal-tide generation and slope steepness, latitude, and stratification
with simulations in an idealized setting and globally in a realistic setting. For this, we use the energy conversion rate (EC) from the barotropic to the baroclinic tide at the model seafloor as a proxy for the generation of internal tides and associated mixing. We then compared the depth of internal-tide generation to the depth at which CWCs occur from a global database of CWC occurrences.

### 3.1 Idealized simulations

We used the parameter settings listed in section 2.3.1. With increasing mean slope steepness, EC rates at the seafloor intensify (Fig. 5a). In our reference setting, for slopes with a steepness <0.03, the depth of maximum EC increases with increasing slope steepness and for slopes with a steepness >0.06, the depth of maximum EC decreases with increasing slope steepness from about 1 km to 0.3 km depth. Regarding latitude, the depth of maximum EC increases from about 0.25 km depth near the poles to about 0.6 km depth at the equator (Fig. 5b). EC rates intensify towards the equator regardless of a change in stratification
(Fig. 5b),and intensify with increasing stratification (Fig. 5c). The depth of maximum EC decreases with increasing stratification from about 0.9 km depth at a stratification of $1.5\cdot10^{-3}$ rad s$^{-1}$ to 0.3 km depth at a stratification of $5\cdot10^{-3}$ rad s$^{-1}$.

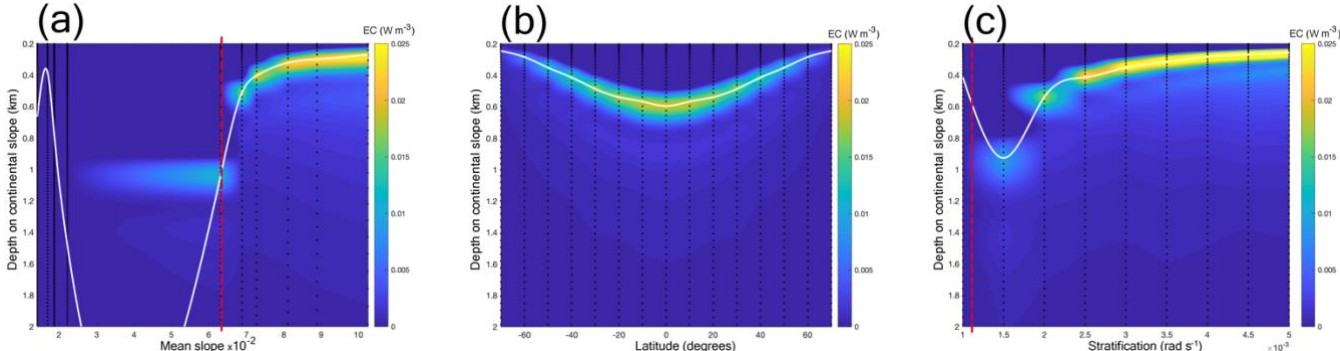

**Figure 5. Panels depict energy conversion (EC) at the seafloor of the continental slope (colours) for different mean slope steepness (a), latitude (b), and stratification (*N*) in the pycnocline (c). The white line depicts an interpolation between the points of maximum EC. Dashed red lines (in a and c) depict the parameter combinations at which the angle of the topography equals the angle of the internal tide beam ('critical' steepness). Black dashed lines indicate the parameter values for which we carried out simulations. In panel a, we plotted the mean slope (as calculated by Eq. (6)), for easy comparison with the realistic simulations, causing a gap. Note that the y-axes begin at 0.2 km depth because the figures depict EC on the continental slope and in our idealized simulation setting the minimum water depth is 200 m.**

### 3.2 Realistic simulation setting

To investigate how the relationships between the depth of internal-tide generation and slope steepness, latitude, and stratification turn out to be in a realistic setting, we simulated internal-tide generation using 84 continental slope transects (Fig. 4) with realistic seasonal stratification. We plotted the average EC at the model seafloor based on available data on depth, slope steepness, and latitude at the transects (Fig. 6), as in Fig. 5. With the idealized topography in the idealized simulation setting only one peak in EC was present, but in the realistic simulation setting multiple peaks in EC were often found. Since CWCs might benefit from a local peak in EC regardless of whether it is the largest peak on the continental slope, we identified all peaks in EC on transects and included them in Fig. 6. We further plotted the weighted mean of depth, slope, and latitude of internal-tide generation for all oceans and seasons together, along with a smoothing spline (Fig. 7a-b). We also plotted the depth at which corals occur (Fig. 7c-d), which will be discussed in the next section (3.3).

The depth at which EC peaks occur increases with increasing slope steepness (Fig. 6a) and internal-tide generation generally occurs about 1 km deeper on slopes with a steepness of 0.05 than on near flat topography (Fig. 7a). In the realistic simulation setting, slope regions with a steepness >0.05 were unusual (only 1.1 % of all slope regions) and most EC peaks were on slope regions with a steepness <0.06. From the poles towards the equator EC intensifies and the depth at which EC peaks occur increases (Fig. 6b). Mean EC depth is shallowest near the poles and around 20 degrees North (~0.5 km depth), increases to about 0.9 km depth near 40 degrees North and 30 degrees South and slightly decreases to 0.7 km depth near 20 degrees South and at the equator (Fig. 7b). A seasonal effect is not evident, so we only display the results for winter in Fig. 6.

These relationships between the depth of optimal internal-tide generation and slope steepness and latitude (i.e., two of the three main factors determining that depth of maximum generation) allow for a comparison to the depth of coral occurrences.

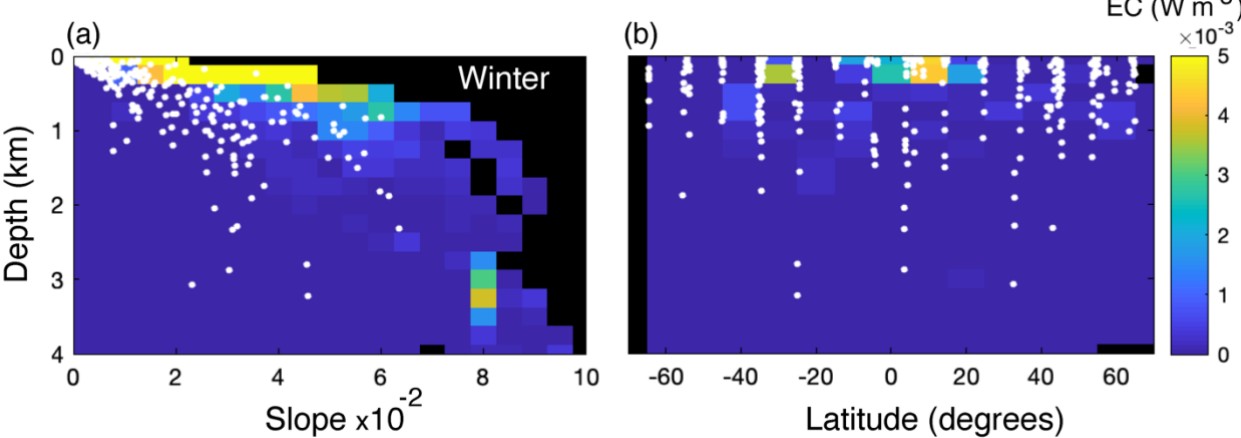

**Figure 6. Colours depict average simulated energy conversion rates at the seafloor (W m$^{-3}$) for different values of slope steepness (a) and latitude (b) from the realistic simulations. The results of simulations with stratification profiles typical of Northern Hemisphere winter are shown since all seasons were similar. Black areas denote parameter combinations that were not present in any of the simulated transects. White dots show peaks in scaled EC.**

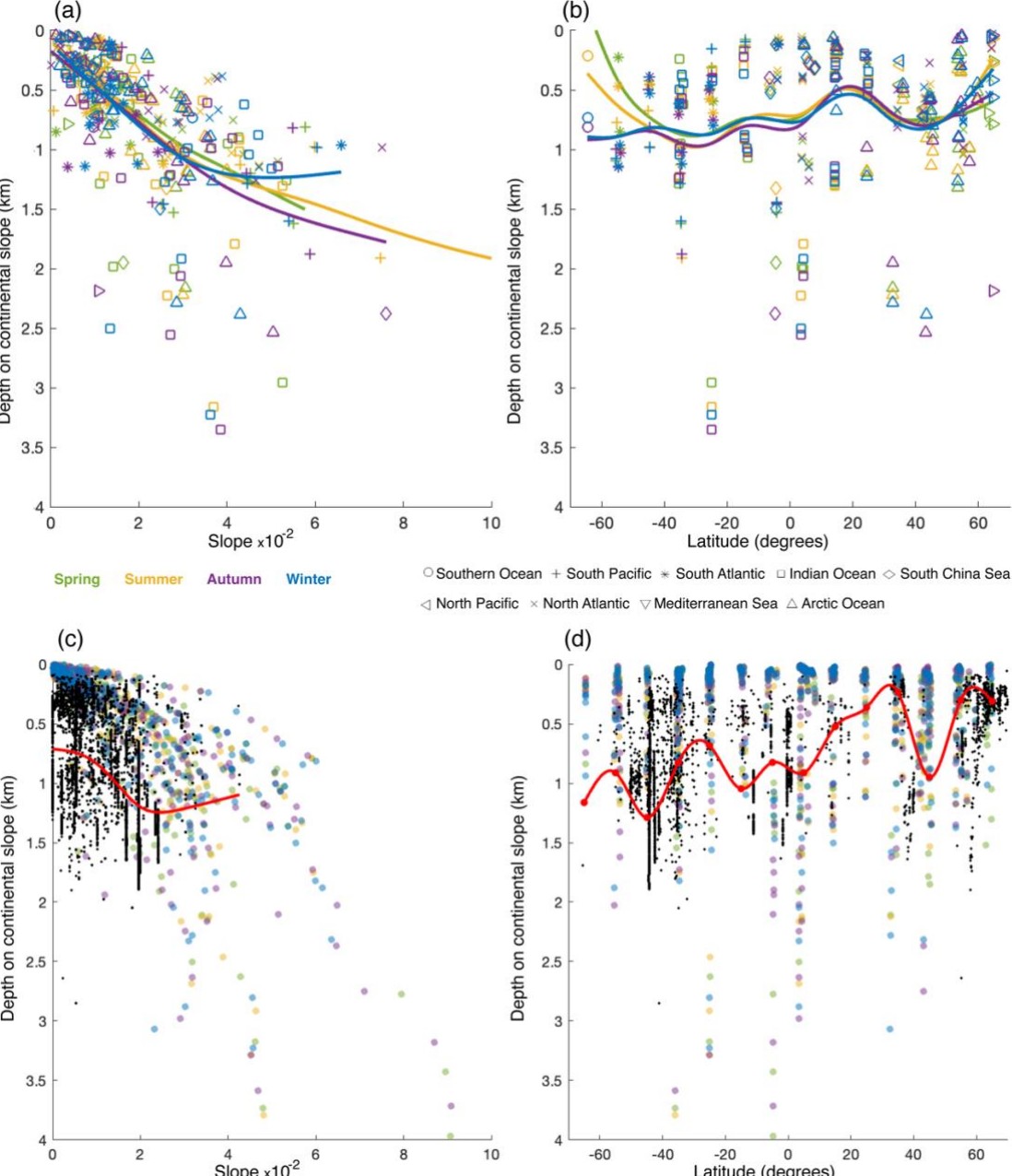

Spring  Summer  Autumn  Winter

○ Southern Ocean  + South Pacific  ✳ South Atlantic  □ Indian Ocean  ◇ South China Sea
◁ North Pacific  × North Atlantic  ▽ Mediterranean Sea  △ Arctic Ocean

**Figure 7. (a) Weighted mean depth of internal-tide generation against weighted mean slope steepness. (b) Weighted mean depth of internal-tide generation against weighted mean latitude. Colours depict the different seasons and symbols the different oceans. Lines depict a smoothing spline through the data from Northern Hemisphere spring (green), summer (yellow), autumn (purple), and winter (blue). (c) Depth at which cold-water corals occur (black dots) against slope steepness. The red line is a smoothing spline through the coral data to indicate the general trend. (d) relationship between the depth at which cold-water corals occur and latitude. The red line is a smoothing spline through a moving median (red dots), showing the general trend. A smoothing spline was fit through a moving average because it failed to fit through the many observations at similar latitudes. Coloured dots in the background represent the EC peaks (as in Fig. 6), where the colours indicate the NH seasons. Coral occurrences are only included from regions where the tide is semidiurnal or mixed but mainly semidiurnal (as in Fig. 2).**

**3.3 Cold-water coral (CWC) occurrences**

To investigate the effect of internal-tide generation on the occurrence of CWCs, we used several global databases with CWC occurrences.

With increasing slope steepness, the depth at which CWCs occur increases, especially for slopes with a steepness <0.03, from about 0.7 km depth to 1.3 km depth (Fig. 7c). The depth at which CWCs occur is shallowest towards the northern pole and around 30 degrees North (~0.3 km depth) and deepest around 40 degrees South (~1.2 km depth; Fig. 7d). The depth at which

corals occur decreases to about 0.7 km depth around 30 degrees South and increases towards 1 km depth near the equator. There is a positive correlation ($r=0.54$, $p<8\cdot10^{-6}$) between the curve for the depth of internal-tide generation against slope steepness (Fig. 7a) and the same curve for CWCs (Fig. 7c), averaged over all seasons. For the correlation coefficients per season see table A1 (Appendix A). The curve for the depth of internal-tide generation against latitude (Fig. 7b) and the same curve for CWCs (Fig. 7d) is most strongly correlated in NH Autumn ($r=0.70$, $p<4\cdot10^{-22}$) and NH Winter ($r=0.65$, $p<2\cdot10^{-18}$),

weakly correlated in NH summer ($r=0.24$, $p<5\cdot10^{-3}$), and negatively correlated in NH spring ($r=-0.27$, $p<2\cdot10^{-3}$). This indicates that the depth-pattern of CWC occurrences is very similar to the depth-pattern of internal-tide generation in NH Autumn and NH Winter.

Besides a general comparison globally, we also compared the depth of peaks in internal-tide generation and the occurrence of CWCs on the 17 transects where CWCs were situated on or very nearby the transect (Fig. 8). We included only those transects

in regions where the tide is semidiurnal or mixed but mainly semidiurnal and used scaled EC since CWCs might benefit from a peak in EC regardless of the intensity of that peak. On transects 6, 9, 11, 13, 18, 20, 90, 105, 114, 115, and 116, the depth at which CWCs are found coincides with the depth of a peak in EC. On transects 33, 51, 90, and 91 corals occur within 200 m depth of increased EC as simulated by our model and on transects 17, 18, 52, and 106 there are corals occurring >600 m depth from increased EC. Cold-water corals thus occur near peaks in internal-tide generation in 11 out of the 17 transects, but the

relation between CWCs and internal-tide generation as found on a global scale is not necessarily indicative for coral occurrences at a regional level.

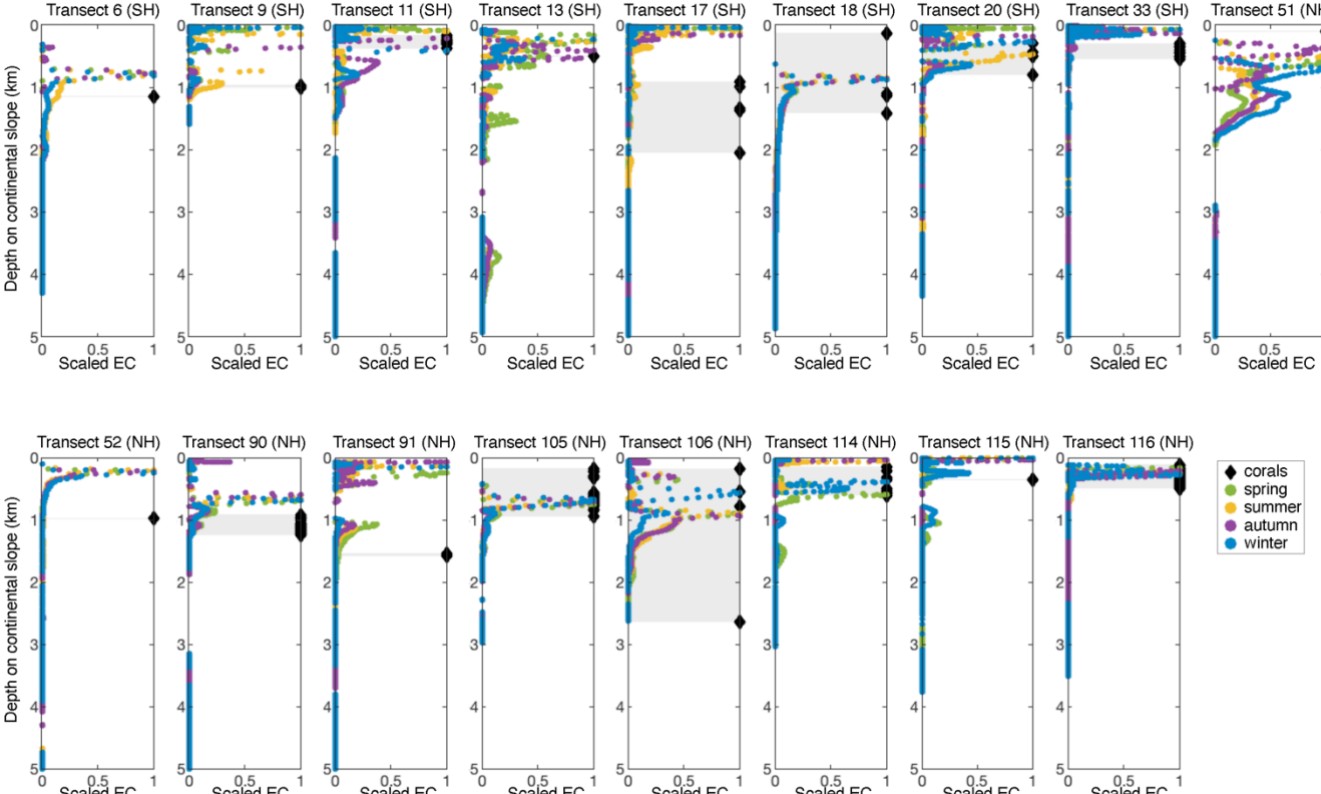

**Figure 8. Scaled energy conversion to the internal tide (EC) on the model seafloor of 17 transects with overlapping cold-water coral occurrences in regions where the tide is semidiurnal or mixed but mainly semidiurnal (as in Fig. 2). Titles indicate the transect number and whether it is located on the Southern Hemisphere (SH) or Northern Hemisphere (NH). Shaded areas indicate the region connecting coral occurrences. Note that the season indication refers to NH seasons.**

### 3.4 Critical reflection and trapped internal tides

At any part of a continental slope there are four possible interactions between the topography and the internal tide. Internal tides typically reflect from the topography and travel away to the open ocean. But, above the critical latitude of around 75 degrees for the M2 tide and 30 degrees for the K1 tide, the internal tide is trapped at the topography. Below the critical latitude a region on the continental slope can be steeper than the internal tide beam (supercritical), as steep as the internal tide beam (critical), or gentler as the internal tide beam (subcritical).

Our realistic transects were located between 65 degrees South and North, so there were no regions on the transects where the M2 internal tide was trapped at the topography. The K1 internal tide was trapped at the topography on 54.7% of all regions on the realistic transects (Fig. 9 and Table 1). The proportion of CWC locations with trapped K1 internal tides (67.7%) is significantly higher than on the transects (Table 1). On all transects, the topography is supercritical and critical for the M2 tide in 1.0% and 3.0% resp. of all regions. The proportion of supercritical and critical topography is significantly higher in the regions where we identified peaks in scaled energy conversion rates (4.5% and 14.5% resp.) and at CWC locations (2.7% and

3.7% resp.). The topography was supercritical and critical for the K1 tide in 1.2% and 6.6% resp. of all regions on all realistic

transects. The percentage of supercritical reflection of the K1 internal tide was significantly higher in CWC locations (11.2%).

**Table 1. The percentages of topography where the M2 (top) and K1 (bottom) internal tide is trapped, or where the reflection is supercritical, critical, or subcritical. We calculated the percentages for all parts on all transects ('all transects'), for the transect regions where internal tide generation peaks ('peaks'), for the coral locations using 30 arc-minute bathymetry ('Coral locations 30'), for coral locations using 1 arc-minute bathymetry ('Coral locations 1'), and for the global ocean using bathymetry at 1 arc-minute resolution ('Globally 1'). The mean and 95% confidence interval are obtained by bootstrapping with 1,000 repetitions.**

| | M2 | | | | | | | | | |
| --- | --- | --- | --- | --- | --- | --- | --- | --- | --- | --- |
| | All transects | | Peaks | | Coral locations 30 | | Coral locations 1 | | Globally 1 | |
| | mean | 95% | mean | 95% | mean | 95% | mean | 95% | mean | 95% |
| Trapped | | | | | | | | | | |
| Super-critical | 1.0 | 1.0 – 1.0 | 4.5 | 3.5 – 5.5 | 2.7 | 2.6 – 2.8 | 66.9 | 66.2 – 67.5 | 9.4 | 9.4 – 9.4 |
| Critical | 3.0 | 3.0 – 3.0 | 14.5 | 12.7 – 16.3 | 3.7 | 3.5 – 3.8 | 15.5 | 15.2 – 15.9 | 3.6 | 3.6 – 3.6 |
| Sub-critical | 96.0 | 95.9 – 96.0 | 81.0 | 79.0 – 83.0 | 93.6 | 93.1 – 94.2 | 17.6 | 17.2 – 18.0 | 87.0 | 86.9 – 87.0 |
| | K1 | | | | | | | | | |
| Trapped | 54.7 | 54.6 – 54.8 | | | 67.7 | 67.3 – 68.2 | 52.5 | 52.0 – 53.0 | 62.5 | 62.5 – 62.5 |
| Super-critical | 1.2 | 1.2 – 1.3 | | | 11.2 | 11.0 – 11.4 | 45.8 | 45.2 – 46.3 | 17.8 | 17.8 – 17.9 |
| Critical | 6.6 | 6.5 – 6.6 | | | 5.9 | 5.7 – 6.1 | 1.2 | 1.1 – 1.3 | 7.0 | 7.0 – 7.1 |
| Sub-critical | 37.5 | 37.4 – 37.6 | | | 15.2 | 14.9 – 15.4 | 0.5 | 0.4 – 0.6 | 12.6 | 12.6 – 12.6 |

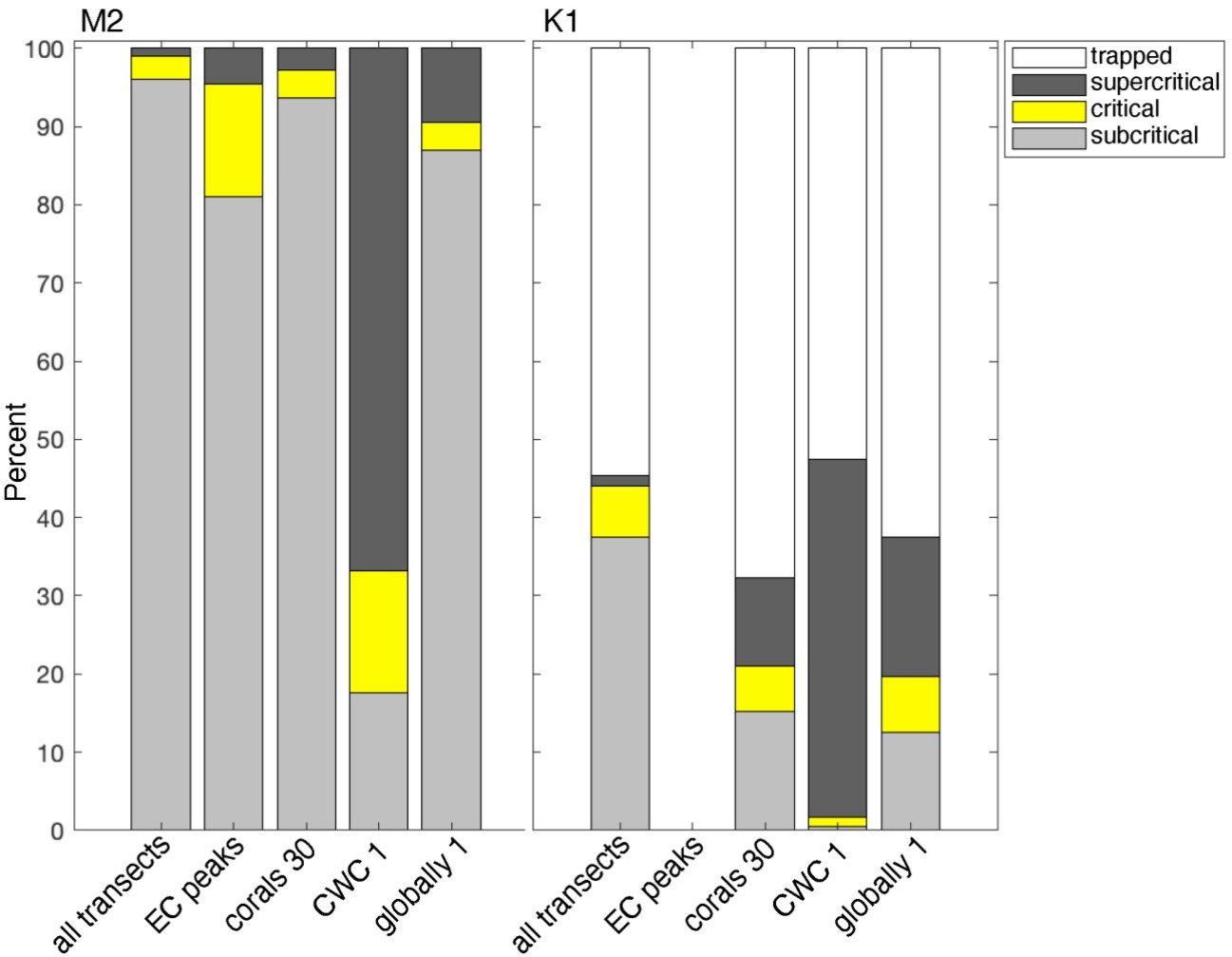

**Figure 9. Bars show the percentage of regions on continental slopes where the internal tide is trapped at the topography (white sections), supercritical (dark grey), critical (yellow), or subcritical (light grey), for internal tides at the M2 tidal frequency (left panel) or K1 tidal frequency (right panel). From left to right, the bars show the percentages of all parts on all transects, slope regions with EC peaks, locations of cold-water coral occurrences calculated with 30 arc-minute topography ('CWC 30'), locations of cold-water corals calculated with 1 arc-minute topography ('CWC 1'), and 10% of the global ocean calculated with 1 arc-minute topography ('globally 1'). We defined 'critical' as a region on the continental slope where the steepness of the topographic slope equals the angle of the internal tidal beam $\pm 5\cdot10^{-7}$. When the Coriolis frequency equals the tidal frequency, the internal tide does not propagate, i.e., is trapped at the topography. The slope of the internal-tide beam depends on season, so these proportions are calculated using all four seasons separately.**

## 4. Discussion

### 4.1 Study results and limitations

We found similarities between the global depth-pattern of internal-tide generation on continental slopes and the depths of CWCs occurrences globally. The relation between internal tides and CWCs is most obvious in the Northern Hemisphere, likely

because more CWC observations are available there. The relation is also strongest for NH Autumn and NH Winter which might reflect a dependency on hydrodynamically mediated food supply mechanisms in food-limited winter months (van der Kaaden et al., 2021; Maier et al., 2020).

By focussing on internal-tide generation at continental slopes with smoothed (low-resolution) bathymetry we investigated how the depth of internal-tide generation changes generally with continental slope steepness and latitude. Our model limitations include the use of a rather coarse grid for the seasonal stratification, which is justifiable for our broad-scale approach, but might cause some deviations in our calculations of the proportions of slope criticality with high-resolution bathymetry. We further used a relatively narrow band (i.e., $\pm 5 \cdot 10^{-7}$) to calculate topographic slopes that are critical for the internal tide, so some slope regions that we defined as subcritical or supercritical might be closer to critical conditions in terms of the hydrodynamics on site. The CWC database further shows a large sampling bias. We tackled this problem by using the median depth for latitudes. Another possibility would be to project the median depth of CWCs on a grid with the same resolution as the bathymetry, with a loss of information as a result.

We further did not simulate the open ocean where internal waves can be generated at rough topography on the boundaries of water masses at depths beyond the stratification profiles used in our simulations (Nikurashin and Ferrari, 2013). Cold-water coral occurrences away from the continental slope (Fig. 3) might be associated to such internal waves in the open ocean. However, most coral observations in our study were located within the maximum depth of our stratification profiles, i.e., 1.45 km depth (Fig. 7c-d). So, our simulations capture the most important features of internal-tide generation for CWCs on continental slopes.

The depth of internal-tide generation typically increases from the poles towards the equator (Fig. 5b) but decreases around the equator (Fig. 7) likely because of a relatively shallow water-column stratification. Internal tidal waves propagate on the surface of isopycnals, so enhanced stratification increases the generation of internal tides at the depth of the pycnocline (Gerkema, 2019; Juva et al., 2020; Legg and Klymak, 2008). A shallow seasonal pycnocline thus decreases the mean depth at which internal tides are generated whereas deep stratification increases this depth (Gerkema et al., 2004). So, strong permanent stratification around 200 m depth, between 15 degrees South and 20 degrees North (Appendix A: Fig. A2), likely causes internal tides to be generated at shallower depth on the continental margin, and the stronger seasonal stratification north than south of the equator likely explains why internal-tide generation is shallower north than south (Fig. 7b).

Previous studies highlighted the importance of deep permanent stratification for CWC growth and coral mound development (Rüggeberg et al., 2016; Wienberg et al., 2020; Matos et al., 2017; Wang et al., 2019; White and Dorschel, 2010), e.g., because of enhanced tidal currents in the pycnocline (White and Dorschel, 2010). We corroborate these results here by showing that CWCs globally have a depth-pattern (Fig. 7d) that is similar to the depth-pattern of stratification (Appendix A: Fig. A2) and that this might be related to the depth of internal-tide generation (Fig. 7b).

## 4.2 Internal tides and other food supply mechanisms for cold-water corals (CWCs)

We investigated global depth-patterns, so our results cannot simply be extrapolated to the finer spatial scale of regional studies. Indeed, there is a considerable spread in the depth at which CWCs occur, indicating that, at a regional scale, alternative mechanisms might control CWC occurrence, thereby changing the relation between internal-tide generation and CWCs.

Continental slope steepness has a larger effect on the depth of internal-tide generation than latitude (Fig. 7a-b), which can be one explanation for the large spread in the depths of internal-tide generation and CWC occurrences. Cold-water corals further often occur in deep canyons (e.g., Pearman et al., 2020; Gori et al., 2013; Price et al., 2021) that can be a sink of particulate organic matter by focussing internal tides (Allen and Durrieu de Madron, 2009; Wilson et al., 2015). However, such canyon were not included in our analysis of continental slopes.

Furthermore, the increased energy dissipation from the reflection of internal tides on (super)critical topography especially benefits benthic life (Mohn et al., 2023; van Haren et al., 2014). We show here that CWCs globally occur more often on those continental slope parts that are (super)critical to the M2 tide or supercritical to the K1 tide, than what would be expected based on the percentage of (super)critical topography on all transects (Fig. 9 and Table 1). Several case-studies on continental slopes (Frederiksen et al., 1992; Mohn et al., 2014; Hanz et al., 2019) and in submarine canyons (Pearman et al., 2020, 2023) relate

CWCs to (super)critical topography. Such (super)critical reflection of internal tides can be a more local phenomenon, and, indeed, with higher resolution bathymetry the relationship between CWCs and (super)critical topography is even more pronounced (Fig. 9 and Table 1).

Cold-water corals can benefit from the (super)critical reflection of internal tides because the increased wave action and turbulence increase downward mixing of organic matter and resuspension of the sediment (Hosegood et al., 2004; Lamb, 2014;

Frederiksen et al., 1992; Hanz et al., 2021). (Super)critical reflection has also been associated to the entrapment of organic matter in nepheloid layers (Wilson et al., 2015; Lamb, 2014) that can benefit CWCs by bathing them in water with a large particle load, and has been suggested to stimulate surface primary productivity (Frederiksen et al., 1992; Davies et al., 2009; Hanz et al., 2019), which can benefit CWCs by increasing the organic matter export towards the deep-sea (Maier et al., 2023; da Costa Portilho-Ramos et al., 2022).

Many other site-specific food supply mechanisms have been described that can weaken the association of CWCs to internal tides. For example, high surface productivity has been found as a factor controlling CWC growth (White et al., 2005; Eisele et al., 2011; Fink et al., 2013; Wienberg et al., 2022; Maier et al., 2023) in which case CWCs might be able to survive at greater depths. For CWC reefs within a few hundred meters from the ocean surface (e.g., Norwegian reefs), wind-induced Ekman transport is likely an important food supply mechanism (Thiem et al., 2006). Furthermore, the presence of specific water

masses (Schulz et al., 2020; Dullo et al., 2008) or a certain density envelope (Flögel et al., 2014; Dullo et al., 2008) have been mentioned as environmental drivers of CWC occurrence. Cold-water corals have further been associated to rough topography (e.g., De Clippele et al., 2021; Lo Iacono et al., 2018; Guinan et al., 2009; Dolan et al., 2008) that would not be resolved in our

model, but it can be questioned whether rough topography is an environmental driver of CWC settlement or whether the rough topography is created by the coral reefs themselves (De Clippele et al., 2017, 2021; van der Kaaden et al., in press).

The formation of CWC mounds might also change the relationship between the depth of CWCs and internal-tide generation. Cold-water coral reefs can develop into coral mounds when reef growth and sediment supply are sufficient (van der Land et al., 2014; Wang et al., 2021; Pirlet et al., 2011). Internal (tidal) waves have been associated to the region of CWC mound initiation (van der Kaaden et al., 2021; Wang et al., 2019; Wienberg et al., 2020; De Mol et al., 2002). Most mounds are some tens of meters high (Freiwald, 2002), but they can become several hundred meters high (Wheeler et al., 2007). So, the present

depth at which some CWCs occur (on the mound) might be several hundred meters higher than the depth at which the corals initially settled (on the seafloor). Already from some tens of meters high, CWC mounds exert an effect on their environment, likely increasing the food supply towards the coral reefs and possibly surmounting the environmental control from ambient environmental processes such as internal-tide generation (van der Kaaden et al., 2021; Soetaert et al., 2016). We hypothesize that the depth of internal-tide generation is important for allowing initial CWC settlement in those continental slope regions

where internal tides are generated.

## 5. Conclusion and outlook

Cold-water coral (CWC) reefs are highly productive ecosystems in the deep-sea that benefit from mechanisms enhancing the vertical transport of high-quality organic matter from near the ocean surface (da Costa Portilho-Ramos et al., 2022; Snelgrove et al., 2018; Cathalot et al., 2015). Internal tides are beneficial for CWC reef and mound formation because they boost the

vertical transport of organic matter from near the ocean surface towards the seafloor (de Froe et al., 2022; Mohn et al., 2014; Frederiksen et al., 1992; Davies et al., 2009). Previous studies suggested that the region on the continental slope where internal tides are generated is especially important for CWCs (van der Kaaden et al., 2021). We found that the global depth-pattern of internal-tide generation against latitude correlates to the depth-pattern of CWCs, especially in NH Autumn and NH Winter, underlining the relation between internal tides and CWCs.

Our study provides insight into the global depth-pattern of internal-tide generation and CWCs, addressing the relation between several general features (i.e., internal tides and stratification) and broad-scale distribution patterns of CWCs. At a regional scale, the mean depth of internal-tide generation on continental slopes might not be the best predictor of individual CWC occurrences as internal tides can be generated at multiple depths along a continental slope and alternative food supply mechanisms to CWCs exist that might make the corals less dependent on internal tidal waves. Nonetheless, we think that it is

interesting to consider the depth of internal-tide generation as a process fostering CWC (mound/reef) growth (van der Kaaden et al., 2021) and as a parameter in habitat suitability modelling (Pearman et al., 2020; Mohn et al., 2023). Furthermore, we showed that CWCs are significantly more often than randomly situated on topography that is (super)critical to the internal tide. We further presented the global relationships between the depth of internal-tide generation on continental slopes and the three parameters governing internal-tide generation. From the poles towards the equator internal-tide generation typically deepens,

but strong seasonal stratification around 25 degrees North and South and strong permanent stratification near the equator

decrease the depth at which internal tides are generated. Slope steepness has a larger effect on the depth of internal-tide generation than latitude, but the effect of latitude is considerable.

Climate change might increase stratification globally (Reid et al., 2009; Li et al., 2018; Capotondi et al., 2012), which will likely increase the energy contained in (Yadidya and Rao, 2022) and the mixing induced by internal tides (Haigh et al., 2019).

Based on our study, global warming might also cause internal tides to be generated shallower on continental slopes. So, with a warming of the climate, new suitable habitat for CWCs might be created shallower on continental slopes.

Since warmer temperatures increase the energy demand of CWCs (Dodds et al., 2007; Dorey et al., 2020; Chapron et al., 2021), suitable CWC habitat is expected to deepen in a warmer climate (Morato et al., 2020). A sufficient food supply to CWCs can however compensate adverse environmental conditions to some degree (da Costa Portilho-Ramos et al., 2022;

Hebbeln et al., 2020; Dorey et al., 2020; Büscher et al., 2017). Wienberg et al. (2020) similarly found (for the Belgica cold-water coral mound province) that the depth of CWC growth decreased, following a decrease in the depth of internal wave activity, which they linked to water mass boundaries. Since at shallower depths coral food supply is higher, the creation of new suitable habitat at shallower regions on continental slopes might provide a mechanism whereby CWCs can compensate the higher energetic costs from warmer temperatures.

**Appendix A**

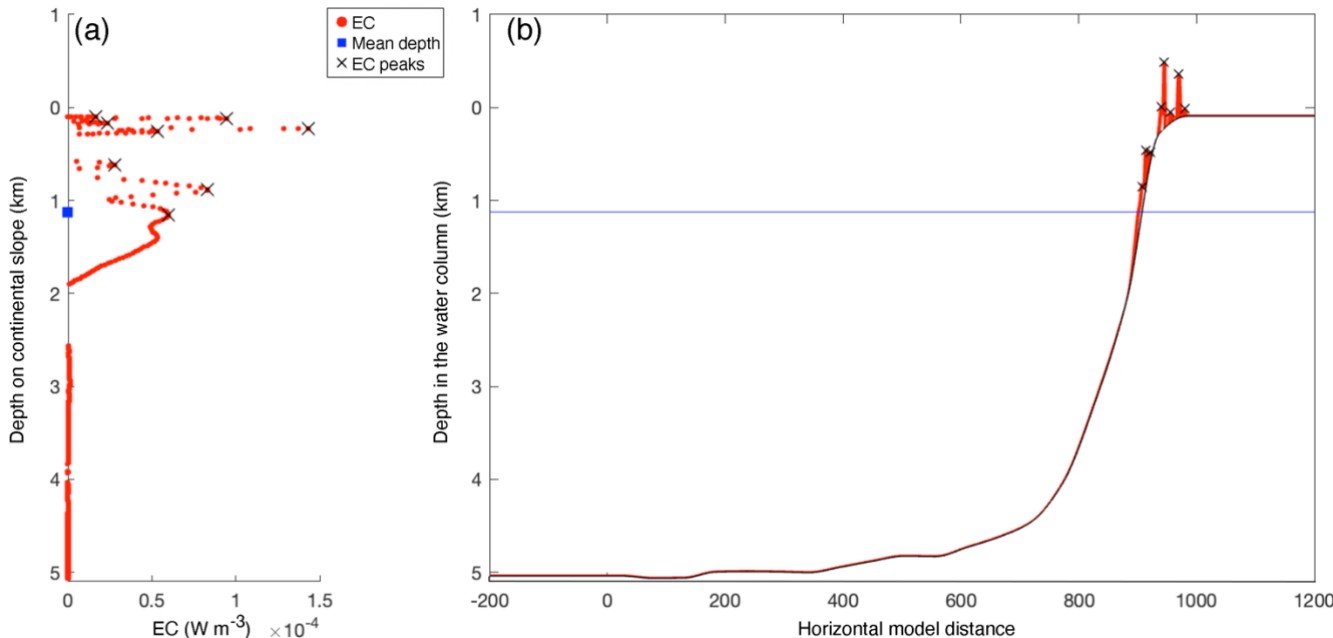

**Figure A1. Example of energy conversion rates (EC) at the model seafloor on transect 12 (the 'winter' simulation), with peaks as identified by Matlabs peaks-finding algorithm and weighted mean depth as calculated by Eq. (6). (a) energy conversion rate (red) on the continental slope. Black crosses indicate peaks in EC and the blue square indicates the weighted mean depth as calculated by**
**Eq. (6). The y-axis denotes the depth on the continental slope, i.e., $z$ in Eq. (6). (b) energy conversion rate (red) plotted on the transect**

topography (black). Black crosses indicate peaks in EC and the blue line indicates the weighted mean depth. The y-axis denotes the depth through the water column.

**Table A1. Pearson correlation coefficients for the correlation between the depth-pattern of internal-tide generation and cold-water coral occurrences. Correlations for the curves of depth against slope steepness (Fig. 7a&c) and latitude (Fig. 7b&c) are shown. Significant correlations ($p<0.01$) are marked by an asterisk.**

|  | Spring | Summer | Autumn | Winter |
|---|---|---|---|---|
| Slope steepness ($r$) | 0.45* | 0.56* | 0.48* | 0.67* |
| $p <$ | $3 \cdot 10\text{-}4$ | $4 \cdot 10\text{-}6$ | $9 \cdot 10\text{-}5$ | $5 \cdot 10\text{-}9$ |
| Latitude ($r$) | -0.27* | 0.24* | 0.70* | 0.65* |
| $p <$ | $2 \cdot 10\text{-}3$ | $5 \cdot 10\text{-}3$ | $4 \cdot 10\text{-}22$ | $2 \cdot 10\text{-}18$ |

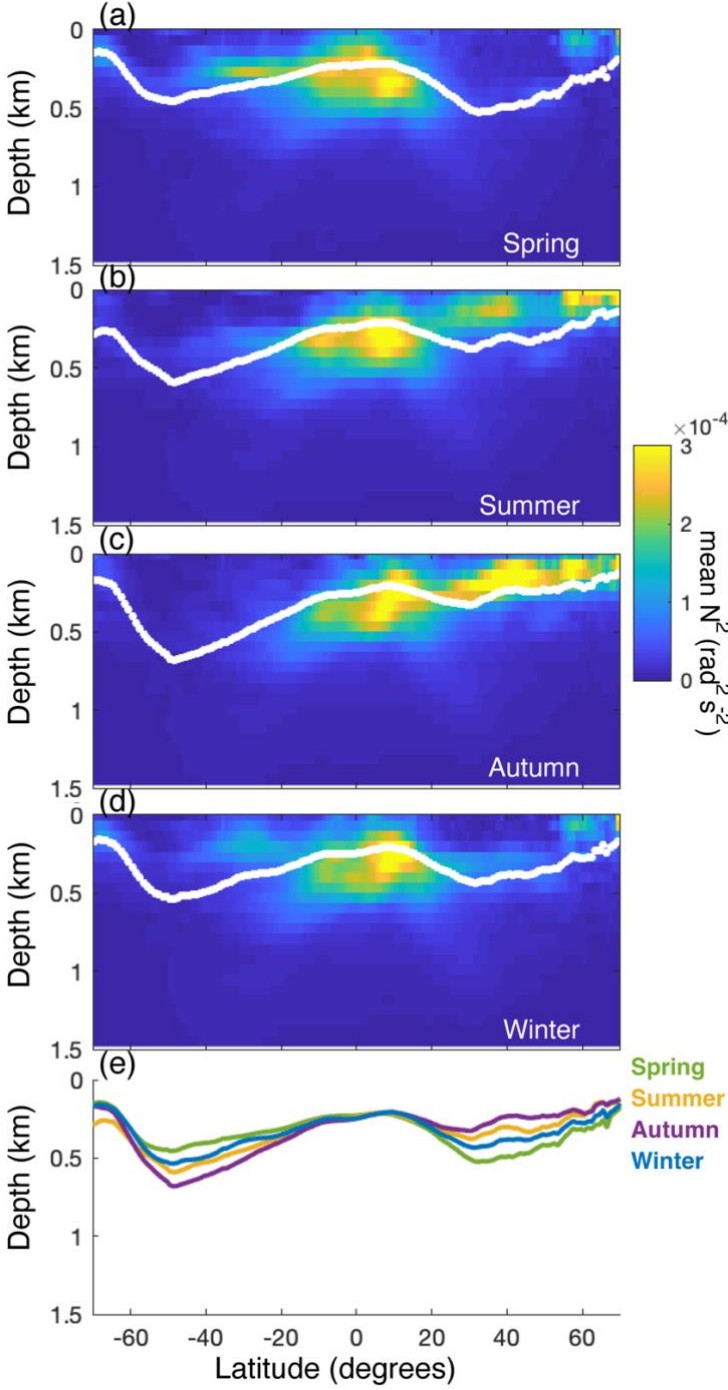

Figure A2. Intensity of stratification ($N^2$; calculated from the Levitus database) through the water column for every latitudinal degree, averaged over every longitudinal degree for (a) February-April ('spring'), (b) May-July ('summer'), (c) August-October ('autumn'), and (d) November-January ('winter'). White dots depict the weighted mean depth of stratification (Eq. (6)). (e) The relationship of the mean depth of stratification against latitude in spring (green), summer (yellow), autumn (purple), and winter (blue).

**Code availability**

The internal tide model has been described in detail in Gerkema et al. (2004) and succinctly in the methods section of this paper. The model code can be made available on request.

**Data availability**

Data on cold-water coral occurrences were obtained from the following open databases and used as described in the methods section of this paper: the NOAA National Database for Deep-sea Corals and Sponges (NOAA National Database for Deep-Sea Corals and Sponges, version 20220426-0), ICES Vulnerable Marine Ecosystems (International Council for the Exploration of the Sea, June 2022) and OBIS (OBIS, 2022).

**Author contributions**

AvdK, DvO, and TG were involved in the conceptualization of the study. The work was supervised by DvO and TG. The model used was developed by TG. The study was executed, and the results were written by AvdK. DvO, JvdK, KS, and MR secured funding for the project. All co-authors commented on the final draft of the manuscript.

**Competing interests**

The authors declare that they have no conflict of interest.

**Acknowledgements**

We thank Emil Sigmann Engh for constructing the database of cold-water coral occurrences. This research has been made possible with collaboration funding between the Royal Dutch Institute for Sea Research and Utrecht University. CM has received funding from the European Horizon 2020 Research and Innovation Programme under grant agreement no. 818123 (iAtlantic). The output of this study reflects only the author's view, and the European Union cannot be held responsible for any use that may be made of the information contained therein.

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
