# Peer review of "Resemblance of the global depth-distribution of internal-tide generation and cold-water coral occurrences"

_EGUsphere, 2023_

## Author Response (AR1)

Reply to reviewer 1

Dear authors,

Please see below my review of your manuscript entitled "The global correlation between internal-tide generation and the depth-distribution of cold water corals" submitted to EGUsphere. In this manuscript, the authors are trying to link the zone of internal tide generation to the presence of cold-water corals (CWC). Using 2 types of simulation (idealized and realistic), the authors explore where along the global ocean slopes the internal tide energy conversion occur, and how it compares with the distribution of CWC.

I can see that a lot of work had been put into this study (tenss of simulation using different setup and forcings). Unfortunately, the results seem to me inconclusive, at least the way they are presented. Negative results themselves do not preclude, in my view, a publication. I am more worried, however, in the way some results are presented and I cannot recommend  a potential publication of this study in its current state. I would suggest a rejection (with invitation to resubmit) or a major revision.

Dear reviewer 1, thank you for your helpful comments on our manuscript. We provided additional statistics to substantiate our most important hypotheses and we changed the manuscript title so that it reflects better the context of our study. Our aim is to investigate how the depth of internal-tide generation changes generally with respect to continental slope steepness and latitude and how these global patterns relate to occurrences of cold-water corals. In our revised manuscript, we better highlight the limitations of our study and that at the regional scale cold-water coral occurrences might be steered by different processes. We also elaborate on the regional processes that can cause the depth-pattern of cold-water coral occurrences to deviate from the depth-pattern of internal-tide generation. We also worded our conclusions more carefully throughout the manuscript. The additional statistics that we conducted supported our main conclusions.

GENERAL COMMENTS

1. The title is misleading: The global *correlation* between internal-tide generation and the depth-distribution of cold water corals. Never such correlation is presented. Only vague statement such as "CWC were *often* located close to EC peaks" (L.308) are provided, which are qualitative and not quantitative. In addition, statements such as "more than what would be expected by chance" (abstract) and "We here also showed that cold water corals are more often than randomly associated to trapped (diurnal) internal tides" (L.405) are simply wrong:

you did not show that these relationships were statistically significant other than they just co-occur... sometimes.

We agree that the term "correlation" is misleading, so we changed it to "Resemblance of the global depth-distribution of internal-tide generation and cold-water coral occurrences." We think that this title covers better the most important aspect of this manuscript.

We like very much the suggestion to perform additional statistics to substantiate our conclusions. For the revised manuscript, we calculated the correlation between the curves for the depth of internal-tide generation and cold-water corals to slope steepness and latitude

(Fig. 7 and Table A1). The depth-pattern of internal-tide generation against slope steepness correlates to the same depth-pattern for cold-water corals (r=0.54, p<0.05), averaged over all seasons. The latitudinal depth-pattern of cold-water corals significantly (p<0.05) correlates to the depth-pattern of internal-tide generation in summer (r=0.24), august (r=0.70), and winter (r=0.65) and negatively in spring (r=-0.27).

For the proportion of supercritical, critical, subcritical, or trapped internal tides we calculated 95% confidence intervals by bootstrapping 1,000 repetitions. We further added the proportions of supercritical, critical, subcritical, and trapped internal tides for the global ocean and cold-water coral locations using higher resolution (1 arc-minute topography from ETOPO1). These confidence intervals are presented in table 1. Comparing the percentage of trapped K1 internal tides at cold-water coral locations to the global ocean, we did not find that cold-water corals occurred more often than by chance at locations of trapped internal tides, so we removed this aspect from our manuscript.

The statement in line 308 that "cold-water corals were often located close to EC peaks" is indeed a bit vague, so we removed it.

Instead, we now state (line 21): "We further found that cold-water corals are significantly more often situated on a topography that is steeper than the internal-tide beam (i.e., where supercritical reflection of internal tides occurs) than can be expected from a random distribution: In 66.9% of all cases, cold-water corals occurred on a topography that is supercritical to the M2 tide, whereas globally only 9.4% of all topography is supercritical." And (line 441): "Furthermore, we showed that cold-water corals are significantly more often than randomly situated on topography that is (super)critical to the internal tide." These claims are confirmed by the added statistics and using higher resolution bathymetry.

2. Some statistics are presented (e.g. Fig. 9 and accompanying text). The problems is that these stats are based on hand-picked transects, and it is unclear how these transects are representative of the global ocean. Thus statements such as "cold-water corals also occur more often at locations where the K1 internal tide is trapped at the topography (67.9 %; Fig. 9), than what would be expected based on the percentage of trapped K1 internal tide on all transects (54.7 %)" is not necessarily representative of the global ocean and may well be just a fluke. The authors could partly get away (or at least estimate the representativeness of the subsample) by randomly resample their data using, for example, bootstrap methods. That would give at least a range of uncertainty/spread.

We agree that it would be better to present a confidence interval for these percentages, so we followed your suggestion for bootstrapping (resulting in table 1).

The locations of the transects were in fact not handpicked. We drew transects along all coastlines every 10th degree between 65 degrees South and North. We had to exclude some transect simulations, especially along the East Pacific coast, but given the even spread in the rest of the oceans, we still consider these transects to be representative of continental slopes globally. We clarified the procedure in the methods section 2.3.2 (line 183): "For the realistic simulation setting, we selected transects on continental slopes from all ocean basins at every 10th latitudinal degree between 64.5 degrees South and 64.5 degrees North, resulting in 116 transects."

We also included slope criticality for cold-water corals and the global ocean using a higher spatial resolution bathymetry (1 arc-minute), to make the comparison more representative of the global ocean. Furthermore, in the discussion we present our results as representative of continental slopes globally and mention specifically that our study does not include internal wave generation in the open ocean.

We highlighted this even more clearly in the revised manuscript (line 355): "By focussing on internal-tide generation at continental slopes with smoothed (low-resolution) bathymetry we investigated how the depth of internal-tide generation changes generally with continental slope steepness and latitude. […] We further did not simulate the open ocean where internal waves can be generated at rough topography on the boundaries of water masses at depths beyond the stratification profiles used in our simulations (Nikurashin and Ferrari, 2013). CWC occurrences away from the continental slope (Fig. 3) might be associated to such internal waves in the open ocean. However, most coral observations in our study were located within the maximum depth of our stratification profiles, i.e., 1.45 km depth (Fig. 7c-d). So, our simulations capture the most important features of internal-tide generation for CWCs on continental slopes."

And (line 383): "We investigated global depth-patterns, so our results cannot simply be extrapolated to the finer spatial scale of regional studies. Indeed, there is a considerable spread in the depth at which CWCs occur, indicating that, at a regional scale, alternative mechanisms might control CWC occurrence, thereby changing the relation between internal-tide generation and CWCs. Continental slope steepness has a larger effect on the depth of internal-tide generation than latitude (Fig. 7a-b), which can be one explanation for the large spread in the depths of internal-tide generation and CWC occurrences. CWCs further often occur in deep canyons (e.g., Pearman et al., 2020; Gori et al., 2013; Price et al., 2021) that can be a sink of particulate organic matter by focussing internal tides (Allen and Durrieu de Madron, 2009; Wilson et al., 2015)."

3. From the title, I was expecting a global correlation between CWC and internal tide critical slopes. The authors have all they need (global N2, global bathymetry/slope, and global known CWC location), so for each CWC location they can find if the slope becomes critical; and if so at which depth. Was this ever done here or in another paper? I think that such analysis would be quite interesting and be more representative of the global ocean.

We agree that this was misleading and that additional statistics were warranted. We considered the approach that you suggested, but did not implement it because of several problems. At (super)critical topography, internal tides are amplified, but internal tides can be generated also on subcritical topography. In fact, we found that peaks in internal tide generation occur more often at locations where the topography is (super)critical than what would be expected based on the percentage of (super)critical topography on all transects, but the largest proportion of topography at internal-tide-peaks is subcritical (table 1). Cold-water corals might benefit also from the dynamics of non-amplified internal tidal waves on sub-critical topography. So, it is not only the criticality of the topography that we are interested in, but really the generation of internal tides. Furthermore, the global distribution of cold-water coral observations is very uneven, which would make the results probably even less representative of continental slopes globally.

Instead, to have a better representation of the global ocean, we calculated the percentages of subcritical, critical, and supercritical topography also for 10% of randomly selected sites globally using 1 arc-minute resolution bathymetry and compared it to the percentages for all

cold-water coral sites calculated with 1 arc-minute resolution bathymetry. We further calculated the correlation between the curves for internal-tide generation and cold-water corals.

4. I think that the Section 5 "Conclusion and Outlook" is really un-appropriate. It is written:

a) "We found that the relationship between the depth of internal-tide generation and continental slope steepness and latitude is very similar to that of deep-sea reef building coral occurrences, in their functional dependence on slope steepness and latitude.

To our knowledge, this is the first time that the global relationships between the depth of internal-tide generation and the three parameters governing internal-tide generation have been elucidated and that the connection of internal tides to cold-water coral occurrences has been made on a global scale"

b) "The depth-pattern of internal-tide generation and latitude is *remarkably similar* to the depth-pattern of cold-water corals, and on transects cold-water corals are often found at peaks of internal-tide generation."

c) "We here also showed that cold water corals are more often than randomly associated to trapped (diurnal) internal tides, highlighting trapped internal tides as an interesting potential food supply mechanism to cold-water corals, which deserves further study but falls outside the scope of the model used here."

-> I think that your study does not support any of these statements.

We worded our conclusion more carefully, underlining again that we addressed general, broad-scale processes and that mean depth of internal-tide generation might not be a good predictor at the regional scale.
For example, we write that (line 435): "Our study provides insight into the global depth-pattern of internal-tide generation and CWCs, addressing the relation between several general features (i.e., internal tides and stratification) and broad-scale distribution patterns of CWCs. At a regional scale, the mean depth of internal-tide generation on continental slopes might not be the best predictor of individual CWC occurrences as internal tides can be generated at multiple depths along a continental slope and other food supply mechanisms to CWCs exist that might relax their dependence on internal tidal waves. Nonetheless, we think that it is interesting to consider the depth of internal-tide generation as a process fostering CWC mound growth (van der Kaaden et al., 2021) and as a parameter in habitat suitability modelling (Pearman et al., 2020; Mohn et al., 2023). Furthermore, we showed that CWCs are significantly more often than randomly situated on topography that is (super)critical to the internal tide."

SPECIFIC COMMENTS

- "more than what would be expected by chance" (L.19) -> The ideal wording would be "statistically significant" (but this was not shown)

We now show that the percentages of (super)critical and trapped internal tides are significantly different between all simulated transects and cold-water coral locations. We changed the sentence to (line 21): "We further found that cold-water corals are significantly more often situated on a topography that is steeper than the internal-tide beam (i.e., where supercritical reflection of internal tides occurs) than can be expected from a random distribution: In 66.9% of all cases, cold-water corals occurred on a topography that is supercritical to the M2 tide, whereas globally only 9.4% of all topography is supercritical."

- "We found that, for continental slopes with an average steepness <0.05, the depth of internal-tide generation decreases markedly with increasing slope steepness, with differences of a kilometre or more." (L.288)

-> I don't undertand this sentence.

We removed this sentence and shortened the discussion, focussing more on the limitations of our study.

- Fig. 9: This figure is very difficult to interpret.

We added a description of the construction of Fig. 9 in the methods section 2.4, moved the figure to the results section (3.4) with additional explanation, and provided a table with the 95% confidence intervals for these percentages (Table 1). We hope that this makes it easier to interpret the figure.

- L.310: this is a nice result, but would benefit from better statistics (see comment 2 above).

We agree and added the 95% confidence intervals with bootstrapping, as you suggested (table 1).

- Section 4.2: The title does not reflect the content of that section: The first 2 paragraphs do not talk about CWC, and the 3rd paragraph mention CWC, but in another context.

We agree that the title of this section does not reflect the content. We shortened the discussion on the role of stratification, focussing only on stratification as a food-supply mechanism.

- Figure 10: "The relationship of the weighted mean depth of stratification against latitude..."

-> What is "weighted mean depth of stratification"?

For clarity, we changed the sentences to (line 472): "White dots depict the weighted mean depth of stratification (Eq. (6)). (e) The relationship of the mean depth of stratification against latitude in spring (green), summer (yellow), autumn (purple), and winter (blue)."

- "So, the depth of internal-tide generation therefore most likely determines the depth of cold-water coral mound initiation (van der Kaaden et al., 2021; Wang et al., 2019; Wienberg et al., 2020)"

-> Are these studies showing this? I thought that this research question was the one addressed in your study, but the results are not really conclusive...

The cited studies have linked the initiation of cold-water coral mound formation to the presence of internal (tidal) waves. We link the cold-water coral occurrences (not mounds) to internal tide generation. If the base of a cold-water coral mound is located at the region of internal-tide generation than the corals growing on the mound summit are located at shallower depths.

We agree that the paragraph on cold-water coral mound formation could be clearer, so we changed the paragraph to (line 415):

"Another alternative food supply mechanism for CWCs that can change the relationship between the depth of CWCs and internal-tide generation comes from the formation of CWC mounds. CWC reefs can develop into coral mounds when reef growth and sediment supply are sufficient (van der Land et al., 2014; Wang et al., 2021; Pirlet et al., 2011). Internal (tidal) waves have been associated to the region of CWC mound initiation (van der Kaaden et al., 2021; Wang et al., 2019; Wienberg et al., 2020; De Mol et al., 2002). Most mounds are some tens of meter high (Freiwald, 2002), but they can become several hundred meters high (Wheeler et al., 2007). So, the present depth at which some CWCs occur might be several hundred meters higher than the depth at which they initially settled. Already from some tens of meters high, CWC mounds exert an effect on their environment, likely increasing the food supply towards the coral reefs and possibly surmounting the environmental control from ambient environmental processes such as internal-tide generation (van der Kaaden et al., 2021; Soetaert et al., 2016). We hypothesize that the depth of internal-tide generation is important for allowing initial CWC settlement in those continental slope regions where internal tides are generated."

MINOR COMMENTS

- I would be okay if the authors chose annual stratification instead of seasonal. That could simplify the results.

We removed three seasons from Fig. 6, showing only the results for winter and only consider the average over seasons (i.e., annual stratification) in the correlation between the depth-curves for slope steepness. Since stratification is one of the three parameters affecting the depth of internal-tide generation, we think it is important to keep the seasonal comparison elsewhere.

- L.34: Maybe add "dissolved oxygen" in this list

Indeed, an important parameter that we forgot to mention, thank you.

- L.100: should it be omega instead of sigma?

Yes, thank you.

- L.132: acronym NH not defined.

Yes, indeed. We added the acronyms (line 135): "Note that we classified the seasons with respect to the Northern Hemisphere (NH), so e.g., February to April is NH spring but Southern Hemisphere (SH) autumn."

- Figure 10: units not provided on the colorbar

Thank you. We added the units to the colorbar.

Reply to reviewer 2

Dear authors,

Please find below my comments for your manuscript entitled "The global correlation between internal-tide generation and the depth-distribution of cold-water corals" submitted to EGUsphere.

The authors of this manuscript simulate internal tide generation by modelling and identifying peak areas of internal tide energy conversion using idealized and realistic models, which they then relate to reef forming cold-water coral occurrences.

The concept behind this paper is well thought and the internal tide modelling thorough. However, the methods for comparing distributions between sites if internal tide generation and cold-water coral occurrence could be developed further and would benefit from a statistical analysis as the current plots that are presented show results that are not inclusive. As a result the overall statements about global correlation or across transects made by the authors are not well supported. There are examples where there is a correlation for some of the realistic simulations across transects but it is not universal. Therefore I would recommend the authors consider how they present the results and discussion concerning the correlation between internal tide generation and cold-water coral occurrence and move from global statements to more accurate / regional statements and undertake a broader literature review to better set the work in the context of what has already been done in the field relating cold-water coral occurrences to internal tides and internal tides as mechanisms of food supply as I feel this would strengthen their interpretation of the results. Lasty I would flag that a major reef forming cold-water coral species from the SW Atlantic coinciding with transects 4 and 7 has been omitted from the analysis and its inclusion could influence the results. Plus I would note the influence of the low resolution of the bathymetry used to calculate slope and statements thereafter and the affect of removing complex geomorphological structures from the simulations but not necessarily accounting for this when choosing cold-water coral presences and the bias this introduces when comparing internal tides and cold-water coral occurrences and how it may contribute to variable results.

I would recommend a major revision of the paper as a result of the omission of key reef forming cold-water corals in areas where models are run, generalised statements based on variable results, and a somewhat limited ecological literature review on internal tides and cold-water coral ecology that limits the ability to place the paper in the context of published literature.

I think that this is a very interesting topic and look forward to seeing the maunscript once revised!

Dear reviewer 2, thank you for your positive words and for your helpful comments. We calculated the correlation between the depth-pattern of internal-tide generation and of coldwater corals and added statistics to the percentages of subcritical, critical, and supercritical reflection of internal tides. We used the list of main reef-building coral species given by Freiwald et al. (2004) and Maier et al. (2023), but, indeed, Roberts and Cairns (2014) provide a slightly different list, including *Bathelia candida*. For *Bathelia candida* we found 4 additional eligible datapoints in the NOAA and/or OBIS databse, of which two are near transect 7, but not on representative bathymetry (e.g., with a different orientation). We decided not to add these 4 extra datapoints to the analysis, but we like to point out that from ICES data we selected all 'stony corals' in 'cold-water coral reef', so these might already contain *Bathelia candida* instances. We further added references on the effects of internal tides on benthic life beyond continental slopes and on the physics of internal tides. Thank you for pointing us towards the interesting Pearman-references.

GENERAL COMMENTS

I would recommend that the authors undertake a broader research literature review to better place the findings of their work within the context of what has already been undertaken. For example in L21 – 23 of the abstract states 'The (super)critical reflection of internal tides and trapped internal tides therefore provide an interesting new angle of food supply mechanisms that has not yet been considered in cold-water coral studies'. However, this has been considered by several published papers.

Thank you for pointing this out. We focused on internal tide generation on continental slopes as a food supply mechanism to cold-water corals, but, indeed, the (super)critical reflection of internal tides has been considered more often in canyons and in other benthic communities such as sponge gardens. We added several references to the text (e.g., on lines 55, 378, 388, 399, 406, 418) and changed the abstract to (line 21): "We further found that cold-water corals are significantly more often situated on a topography that is steeper than the internal-tide beam (i.e., where supercritical reflection of internal tides occurs) than can be expected from a random distribution: In 66.9% of all cases, cold-water corals occurred on a topography that is supercritical to the M2 tide, whereas globally only 9.4% of all topography is supercritical. Our findings underline internal-tide generation and the occurrence of supercritical reflection of internal tides as globally important for cold-water coral growth."

The section of the discussion 'Internal tides and other food supply mechanisms for cold-water corals' is very important for explaining the results and feels a little disjointed at present. With some further reading the authors could better present how their work potentially fits into current debates on the role of food supply mechanism and cold-water coral distribution and there by better explain their results.

I would recommend that the authors consider the limitations of their study, especially the use of low resolution bathymetry and filtering out areas of complex geomorphology and how this also affects the results depicted in figure 8.

In the revised manuscript we explain better the value of our study using broad-scale bathymetry in investigating global, general patterns and the limitations, e.g., that we did not simulate in the open ocean, that our stratification profiles do not extent beyond 1.45 km depth, and the coarse resolution of our model topography. We then discuss some of the regional food supply mechanisms in a bit more depth and with additional references.

We shortened the discussion of our results (line 350) and elaborated on the limitations of our study (line 355): "By focussing on internal-tide generation at continental slopes with smoothed (low-resolution) bathymetry we investigated how the depth of internal-tide generation changes generally with continental slope steepness and latitude. Our model limitations include the use of a rather coarse grid for the seasonal stratification, which is justifiable for our broad-scale approach, but might cause some deviations our calculations of the proportions of slope criticality with high-resolution bathymetry. We further used a relatively narrow band (i.e., $\pm 5 \cdot 10^{-7}$) to calculate topographic slopes that are critical for the internal tide, so some slope regions that we defined as subcritical or supercritical might be closer to critical conditions in terms of the hydrodynamics on site. The CWC database further shows a large sampling bias. We tackled this problem by using the median depth for latitudes. Another possibility would be to project the median depth of CWCs on a grid with the same resolution as the bathymetry, with a loss of information as a result.
We further did not simulate the open ocean where internal waves can be generated at rough topography on the boundaries of water masses at depths beyond the stratification profiles used in our simulations (Nikurashin and Ferrari, 2013)."

To address the influence of internal-tide generation at a more regional scale we calculated the slope criticality at a finer topographic scale (Fig. 9). We elaborated on the (super)critical reflection of internal tides as a food supply mechanism in the discussion (line 390).

I would recommend that the authors consider undertaking statistics to test the significance of correlations between internal tide generation and cold-water coral sites as the paper title infers a correlation but the results are inconclusive when considering all the data and descriptive.

We thought this was a very good suggestion, and in agreement with reviewer 1, to add statistics to substantiate our conclusions. We calculated the correlations between the curves for the depth of internal-tide generation and cold-water corals against slope steepness (Fig. 7a&c) and latitude (Fig. 7b&d). The results can be found from line 303: "There is a positive correlation ($r=0.54$, $p<8 \cdot 10^{-6}$) between the curve for the depth of internal-tide generation against slope steepness (Fig. 7a) and the same curve for cold-water corals (Fig. 7c), averaged over all seasons. For the correlation coefficients per season see table A1 (Appendix A). The curve for the depth of internal-tide generation against latitude (Fig. 7b) and the same curve for cold-water corals (Fig. 7d) is most strongly correlated in NH Autumn ($r=0.70$, $p<4 \cdot 10^{-22}$) and Winter ($r=0.65$, $p<2 \cdot 10^{-18}$), weakly correlated in summer ($r=0.24$, $p<5 \cdot 10^{-3}$), and negatively correlated in spring ($r=-0.27$, $p<2 \cdot 10^{-3}$)."

For the proportion of supercritical, critical, subcritical, or trapped internal tides we calculated 95% confidence intervals by bootstrapping 1,000 repetitions. We further added the proportions of supercritical, critical, subcritical, and trapped internal tides for the global ocean and cold-water coral locations using higher resolution (1 arc-minute topography from ETOPO1). These intervals can be found in table 1. Comparing the percentage of trapped K1 internal tides at cold-water coral locations to the global ocean, we did not find that cold-water corals occurred more often than by chance at locations of trapped internal tides, so we removed this aspect from our manuscript.

In general, I find that the use of increasing and decreasing - with increasing depth for shallower depth value miss leading and suggest that they are changed around.

It was not our intention to describe an increase in the depth of internal tide generation by 'decreasing' and vice versa. We corrected all instances where we switched it around. Thank you for noticing this.

To save on words, cold-water corals could be shortened once introduced to CWCs.

This a good suggestion, thank you. We abbreviated most instances.

SPECIFIC COMMENTS

L36 – Benthic organisms rely on other sources of resource such as dissolved organic matter, therefore I recommend the sentence be changed to reflect this.

Indeed, thank you. We removed the generalization of "Benthic organisms", and changed the sentence to (line 40): "Cold-water corals rely on organic matter that ultimately originates from primary production at the sea-surface (Van Engeland et al., 2019, Carlier et al., 2009; Maier et al., 2023)"

L40 – Pearman et al 2020 and 2023 have published work relating CWCs to internal tides.

Thank you for these interesting references. We added Pearman 2023 to the list of studies relating CWCs to internal tides in line 47.

L44-45 Sediment also infills the dead coral framework and is an important component of coral mound formation. Robert et al., 2009.

Yes, we added it (line 48): "Cold-water coral reefs and mounds are built up of dead coral framework, coral rubble, and sediment, often with thriving cold-water coral reefs on the mound tops (Roberts et al., 2009)."

L 79 – should this read 'internal tide generation'?

That is better indeed.

L103 – Please give details or an example of the actual depth resolutions in m.

To the methods section 2.3.2 we added (line 204): "The depth of the shallow and deep parts of the transects ranged from 0.1m to 0.3km and from 1.1km to 8.9km respectively."

L139-141 Here the authors list the cold-water coral species under consideration. Since this is a global study can I recommend that the authors include *Bathelia candida* as it is the main reef-building species of the South-west Atlantic – covered by transects 4 and 7.

We found only four eligible datapoints for *Bathelia candida* that are near transect 7, but not on representative bathymetry (e.g., with a different orientation).

L150 It seems likely that 30 arc- minute is a typo and should be 30 arc-second? If not I would be concerned that the paper makes statements in the discussion that characterises the continental slope as sub- critical and supercritical but from this low resolution data set would not be representative of reality.

We agree that 30 arc-minutes is too broad a resolution to make statements on the criticality of continental slopes globally, so we included a calculation of slope criticality with a 1 arc-minute resolution for the cold-water corals and globally (Fig. 9). Globally most slopes are still subcritical also with higher resolution bathymetry.

L176 – As rough topography represented by canyons are important site of internal tide generation and localities of cold-water corals (cold-water corals show a preference for areas of complex topography) I think that this approach will bias the results. Especially if the authors still include cold- water coral occurrences from these areas of complex topography while at the same time removing this information from the bathymetry by smoothing.

Our approach is to study how the depth of internal-tide generation is affected by continental slope steepness, latitude, and stratification in general. Smoothing the continental slope allowed us to investigate this general relationship. That cold-water corals occur often on rough topography and in canyons might explain part of the large spread in the depth of cold-water coral occurrences. We elaborated on this in the discussion (line 383): "We investigated global depth-patterns, so our results cannot simply be extrapolated to the finer spatial scale of regional studies. Indeed, there is a considerable spread in the depth at which CWCs occur, indicating that, at a regional scale, alternative mechanisms might control CWC occurrence, thereby changing the relation between internal-tide generation and CWCs. Continental slope steepness has a larger effect on the depth of internal-tide generation than latitude (Fig. 7a-b), which can be one explanation for the large spread in the depths of internal-tide generation and CWC occurrences. CWCs further often occur in deep canyons (e.g., Pearman et al., 2020; Gori et al., 2013; Price et al., 2021) that can be a sink of particulate organic matter by focussing internal tides (Allen and Durrieu de Madron, 2009; Wilson et al., 2015)."

L205 The use of the term 'increases' when used to described shallower depths is misleading and I suggest that the use of increasing/decreasing in relation be changed around throughout the text.nFor example, from L207 If the EC depth is getting shallower I would suggest that the depth of maximum EC is decreasing rather than increasing.

Yes, we changed all instances where we switched it around.

L222 – The authors mention the smooth topography used in the simulations and compare it to the realistic simulations – did the authors use different topography smoothing or bathymetry and resolution in the realistic simulations? If so please add these details to the method.

We changed 'smooth' to 'idealized'. The topography in the idealized simulations is perfectly smooth whereas the topography in the realistic simulations is smoothed but not perfectly smooth (see e.g., Fig. A1 for an example of topography used in the realistic simulations).

L228 – The rarity of slopes with steepness >0.05 could merely be an artifact of the low resolution of bathymetry data that the authors are using. Did the authors also omit complex geomorphological features from the realistic simulations from transects? If so the authors are filtering out all the steep slope areas – so the statement is not reflective of reality.

We agree, this is a very good point, so we removed the sentence from the conclusion. Our aim is to address broad-scale, general patterns, for which the smoothed bathymetry works better than bathymetry with all the specific, local features. We calculated the slope criticality also for global topography at a 1 arc-minute resolution (highest ETOPO1 spatial resolution).

The percentage of M2 (super)critical internal tides is indeed higher when calculated with higher resolution topography than with lower resolution topography (Fig. 9), but still less than 15%. Interestingly, for cold-water coral locations the percentage of (super)critical topography is a staggering 82.4% when using fine resolution bathymetry.

L256- 258 Please add a reference for this statement.

This statement (line158) is based on our results as presented in Fig. 3. We refer to figure 3. We moved this figure to the methods section 2.2.2.

L258-259 Please add a reference for this statement.

This statement (line 160) is based on our results as presented in Fig. 2. We refer to figure 2.

Figure 7 captain should read Map with all 'cold-water' coral

Yes, thank you.

L266- the deepest record of cold-water corals may change for the southern hemisphere is *Bathelia candida* is included in the analysis.

The datapoints that we found for *Bathelia candida* are around -42 degrees latitude at a depth of 1.1 km and 1.4 km and around -47 degrees latitude at a depth of 1.2 km and 0.9 km. Hence, adding these datapoints would not change the depth-pattern for cold-water corals meaningfully.

L275 – On review of figure 8, I would interpret the relationship between seasonal EC and coral occurrence as cooccurring with EC peaks on transects 11, 13, 20, 105, 114 and 116 elsewhere I would not make that association. Can the authors run statistics to assess this relationship?

We interpret transect 6, 9, and 115 as an EC peak in winter that coincides with the cold-water coral depth. We did change the wording with respect to transect 33 and 90, saying that cold-water corals occur within 200 m of a peak. Running statistics, we found that mean scaled energy conversion rate is higher at coral locations on these 17 transects than generally on the transects, but we think, even though we scaled the energy conversion rates, it is not a very good approach to compare them directly between transects like this. Figure 8 is more meant as an illustration and we removed references to it from the discussion and conclusion.

L288 -292 The authors emphasis slope steepness values of >0.06 and state that few continental slopes have slope values >0.05 from which they conclude that only a few regions of the continental slope are (super) critical. – I would suggest that the authors consider the resolution of the bathymetry data used in the study and the limitation this imposes on deriving slope values and whether using higher resolution data such as GEBCO would lead to different results. I would highlight that the resolution of the bathymetry data can influence the ability to identify super critical regions eg. Submarine Canyon Walls.

We included an analysis of slope criticality with continental slopes globally using higher resolution bathymetry.

Figure 9. I would suggest changing corals to cold-water corals and explaining K1.

We changed 'corals' to 'cold-water corals' in the caption and 'corals' in the figure to 'CWC' and we changed the first sentence to: "Bars show the percentage of slope regions where the internal tide is trapped at the topography (white sections), supercritical (dark grey), critical (yellow), or subcritical (light grey), for internal tides at the M2 tidal frequency (left panel) or K1 tidal frequency (right panel)."

L310 – change to 'We show here'

OK

L310 – Pearman et al 2020 and 2023 also show relationship between cold-water corals and super critical topography.

We added (line 393): "Several case-studies on continental slopes (Frederiksen et al., 1992; Mohn et al., 2014; Hanz et al., 2019) and in submarine canyons (Pearman et al., 2020, 2023) relate cold-water corals to topography that is (super)critical for the internal tide."

L321-322 – 'Trapped internal tides are thus an interesting hydrodynamic mechanism that likely increases the food supply towards cold-water coral reefs which is not yet often considered in studies on cold-water coral food supply mechanism' This has been considered by several papers and I suggest to undertake a broader literature review.

This relation to trapped internal tides did not show when comparing internal tides at cold-water coral locations to slopes globally, so we removed the part on trapped internal tides from our manuscript.

L352 – Change to 'We show here' and 'as' to 'to'.

OK

L353 – If the authors wish to relate cold-water coral occurrence to depth-stratification could the authors plot cold-water coral occurrence onto Figure 10?

We did not intend to make this comparison. For clarity, we moved figure 10 to the appendix and we shortened the discussion on this topic (line 369).

L363 – The authors statement is contradictory – they state that they show a pattern of co-occurrence between internal tide generation and cold-water coral occurrence and then go on to state that there was however considerable spread. Suggest that the authors either make less general statements and only state transects where there are close associations or do not make general but contradictory statements.

We clarified the difference between our broad-scale approach that allows us to make general statements e.g., about how the depth of internal-tide generation changes with latitude, and the regional scale at which the cold-water coral depth might be determined by another process.

We start the discussion by mentioning the low resolution (line 355): "By focussing on internal-tide generation at continental slopes with smoothed (low-resolution) bathymetry we

investigated how the depth of internal-tide generation changes generally with continental slope steepness and latitude." Then we mention that our results cannot be extrapolated to the regional scale and discuss alternative (more regional) food supply mechanisms (line 383).

L360 – Could reference older papers too that have shown this correlation.

Throughout the manuscript, we now refer to the following studies linking cold-water corals to internal tides: Davies et al. (2009), Frederiksen et al. (1992), de Froe et al. (2022), Hanz et al. (2019), van Haren et al. (2014), Juva et al. (2020), van der Kaaden et al. (2021), Mohn et al. (2014), Mohn et al. (2023), De Mol et al. (2002), Pearman et al. (2020), Pearman et al. (2023), Soetaert et al. (2016), Thiem et al. (2006), Wang et al. (2019)

L365 Please expand on this statement and explain how modified food mechanisms may relate to the broader range of cold-water coral occurrence in relation to EC peaks and include references to support this statement.

We moved this sentence ("Many other site-specific food supply mechanisms have also been described that can weaken the association of cold-water corals to internal tides." on line 405) so that it is the first sentence of the next paragraph instead of the last sentence of the previous paragraph and elaborated on the food supply mechanisms.

L369 – Please explain how trapped internal tides could facilitate cold-water coral growth.. by which mechanism and reference papers to support statement.

We removed the discussion on trapped internal tides.

L398 – Again note that this is based on low resolution bathymetry data, which effects slope values.

This is true, we removed this sentence from the conclusion.

411 – Work has been undertaking looking at this – could be good to include in discussion.

'Shoaling' might not be the best choice of words here. We changed the sentence (line 450) to: "Based on our study, global warming might also cause internal tides to be generated shallower on continental slopes.".
We are not aware of any studies investigating the depth of internal tide generation and how it changes with climate warming. The studies that we found on the effect of global change on internal-tide generation (e.g., Yadidya and Rao, 2022; Haigh et al., 2020) address the *depth-averaged* energy in the internal tides and propagation speed and direction, and not the *depth* of internal-tide generation.

L420 – Increasing the role of internal tides in providing temporal variation in oceanographic variables or temporal relief in anoxic conditions e.g Angola and Whittard Canyon is being considered too.

We think that with the design of our study, we cannot say anything about how internal-tide generation might change in canyons, as the internal-tide dynamics are quite different in canyons as compared to continental slopes. Furthermore, we focused on the cross-slope

direction, with the model assumption of along slope uniformity, while canyons are really a 3D feature.

---

## Referee Report (RR1)

Dear authors,

Please find below my comments for your revised manuscript entitled " Resemblance of the global depth-distribution of internal-tide generation and cold-water Coral occurrences" submitted to EGUsphere.

Thank you for addressing the comments raised in the first review of the paper. The paper now reads more coherently and the results and discussion are improved with wider literature review and the recognition of data limitations and distinguishing global from regional interpretations of findings.

Line – 25 – 'In 66.9% of all cases, cold-water corals occurred on a topography that is supercritical to the M2 tide, whereas globally only 9.4% of all topography is supercritical.'

Could this be changed to 'in our study....66.9% of all cases, cold-water corals occurred on a topography that is supercritical to the M2 tide, whereas globally only 9.4% of all topography is supercritical.'   Or cleary express that this value is expected based on the percentage of all transects.

Line 50 – I would suggest changing 'deeper layers' to 'greater water depths'.

Line 51 – I would suggest changing 'stimulate' to 'facilitate'.

Line 53 – I would suggest changing CWC back to Cold-water coral when it is at the start of a sentence. Same throughout manuscript.

Figures (ALL). Can resolution be improved? Currently, it is not possible to review these figures and their updates.

Line 151 – The authors describe that Bathelia candida was omitted due to few and not in corresponding bathymetry. Since the authors refer to selecting all main reef forming CWCs, and yet make no mention of this prominent reef former of the SW Atlantic they should make a mention of it here to demonstrate that they have considered it and explain why it was omitted or that it may have been included under ICES VME category as they have done in their response.

Line 753 – Should 'cold-water occurrences' by 'CWC occurrences'?

Line 176 – Would it read better as 'Consequently in our analysis'?

Line – 361 – The first section of the results under 'Critical reflection and trapped internal tide' includes introductory sentences on internal tide behaviour – this information could be better placed in the introduction than the results section.

Line 432 – 'might cause some deviations our calculations of the proportions of slope criticality with high-resolution bathymetry'.  Suggest 'might cause some deviations from our calculations of the proportions of slope criticality with high-resolution bathymetry'.

Line 482- The authors state 'CWCs further often occur in deep canyons (e.g., Pearman et al., 2020; Gori et al., 2013; Price et al., 2021) that can be a sink of particulate organic matter by focussing internal tides. (Allen and Durrieu de Madron, 2009; Wilson et al., 2015)' The authors could further exemplify that these features will have been 'smoothed out' from the bathymetry so the reader understand the link.

Line 505 – If linking to the sentence above this sentence could benefit from beginning as ' For example,'

Line 516 – This sentence is unclear.

520 – 'meter' to 'meters'

522- 'might be several hundred meters higher than the depth at which they initially settled.' Do you mean 'might be several hundred meters higher than the depth at which the mound initially established'?

Line 545 – rather than 'relax' it is likely that it makes the relationship more complex?

Line 547 – Add reefs as well as mounds?

Please check the consistency of the use of the term Northern Hemisphere and its acronym NH throughout the text.

---

## Author Response (AR2)

Reply to reviewer

Dear authors,

Please find below my comments for your revised manuscript entitled " Resemblance of the global depth-distribution of internal-tide generation and cold-water Coral occurrences" submitted to EGUsphere.

Thank you for addressing the comments raised in the first review of the paper. The paper now reads more coherently and the results and discussion are improved with wider literature review and the recognition of data limitations and distinguishing global from regional interpretations of findings.

**Dear reviewer, thank you for your comments on our revised manuscript. Below is our detailed reply to your comments. Kind regards, Anna van der Kaaden**

Line – 25 – 'In 66.9% of all cases, cold-water corals occurred on a topography that is supercritical to the M2 tide, whereas globally only 9.4% of all topography is supercritical.' Could this be changed to 'in our study….66.9% of all cases, cold-water corals occurred on a topography that is supercritical to the M2 tide, whereas globally only 9.4% of all topography is supercritical.' Or cleary express that this value is expected based on the percentage of all transects.

**Indeed, it would be better to write "in our study,…". So, we changed it, thank you.**

Line 50 – I would suggest changing 'deeper layers' to 'greater water depths'.

**Indeed, we like your phrasing better, so we changed the sentence to "…, organic matter is degraded by organisms in the water column, decreasing the food quantity and quality for benthic life at greater water depths."**

Line 51 – I would suggest changing 'stimulate' to 'facilitate'.

**We changed it.**

Line 53 – I would suggest changing CWC back to Cold-water coral when it is at the start of a sentence. Same throughout manuscript.

That would be better indeed. We changed CWC to Cold-water coral everywhere where it was at the start of a sentence.

Figures (ALL). Can resolution be improved? Currently, it is not possible to review these figures and their updates.

I will upload the figures in separate files. I noticed that the figures had a very low resolution in the track-changes manuscript, but the right resolution in the revised manuscript (without track-changes).

Line 151 – The authors describe that Bathelia candida was omitted due to few and not in corresponding bathymetry. Since the authors refer to selecting all main reef forming CWCs, and yet make no mention of this prominent reef former of the SW Atlantic they should make a mention of it here to demonstrate that they have considered it and explain why it was omitted or that it may have been included under ICES VME category as they have done in their response.

We added to the methods (line 152): "Roberts and Cairns (2014) also mention the species *Bathelia candida* as a main CWC reef-building species. We did not include this species in our analysis, resulting in the omission of 4 eligible datapoints from the NOAA and/or OBIS databases near transect 7 (SH). The species might be included in the data from ICES, as we selected all "Stony corals" in "Cold-water coral reef" habitat."

Line 753 – Should 'cold-water occurrences' by 'CWC occurrences'?

Yes, thank you.

Line 176 – Would it read better as 'Consequently in our analysis'?

We added 'hence' to the sentence: "…will therefore be limited. Hence, in our analyses, we only included…"

Line – 361 – The first section of the results under 'Critical reflection and trapped internal tide' includes introductory sentences on internal tide behaviour – this information could be better placed in the introduction than the results section.

Yes, this is right and we do explain this in the introduction. However, we noticed that figure 9 (the barplot) was difficult to interpret for readers that were less familiar with the internal tide. We therefore chose to leave the introductory sentences at the beginning of section 3.4 (line 328).

Line 432 – 'might cause some deviations our calculations of the proportions of slope

criticality with high-resolution bathymetry'. Suggest 'might cause some deviations from our calculations of the proportions of slope criticality with high-resolution bathymetry'.

Yes, thank you.

Line 482- The authors state 'CWCs further often occur in deep canyons (e.g., Pearman et al., 2020; Gori et al., 2013; Price et al., 2021) that can be a sink of particulate organic matter by focussing internal tides. (Allen and Durrieu de Madron, 2009; Wilson et al., 2015)' The authors could further exemplify that these features will have been 'smoothed out' from the bathymetry so the reader understand the link.

Yes, thank you, we should elaborate the sentence indeed. It is not that we smoothed them out, but more that we focussed on continental slopes rather than the open ocean.

We changed the sentence to: "Cold-water corals further often occur in deep canyons that can be a sink of particule organic matter by focussing internal tides. However, such canyons were not included in our analysis of continental slopes."

Line 505 – If linking to the sentence above this sentence could benefit from beginning as '

For example,'

Thank you, that is clearer indeed. We changed it.

Line 516 – This sentence is unclear.

We changed the sentence "Another alternative food supply mechanism for CWCs that can change the relationship between the depth of CWCs and internal-tide generation comes from the formation of CWC mounds." to: "The formation of CWC mounds might also change the relationship between the depth of CWCs and internal-tide generation."

520 – 'meter' to 'meters'

Yes, thank you.

522- 'might be several hundred meters higher than the depth at which they initially settled.'

Do you mean 'might be several hundred meters higher than the depth at which the mound

initially established'?

We clarified the sentence to: "So, the present depth at which some CWCs occur (on the mound) might be several hundred meters higher than the depth at which the corals initially settled (on the seafloor)."

Line 545 – rather than 'relax' it is likely that it makes the relationship more complex?

We meant that the corals are less dependent on internal-tide generation for their food supply when they have alternative food supply mechanisms. We clarified the sentence: "…as internal tides can be generated at multiple depths along a continental slope and alternative food supply mechanisms to CWCs exist that might make the corals less dependent on internal tidal waves."

Line 547 – Add reefs as well as mounds?

We changed it to: "CWC (mound/reef) growth."

Please check the consistency of the use of the term Northern Hemisphere and its acronym

NH throughout the text.

We checked whether we used Norther Hemisphere and Southern Hemisphere correctly, and we added NH before the seasons in the text to clarify that we classified the seasons according to the NH not SH.